# Calving-driven fjord dynamics resolved by seafloor fibre sensing

Dominik Gräff[1,2 ✉], Bradley Paul Lipovsky[1], Andreas Vieli[3], Armin Dachauer[3], Rebecca Jackson[4], Daniel Farinotti[2,5], Julia Schmale[6], Jean-Paul Ampuero[7], Eric Berg[8], Anke Dannowski[9], Andrea Kneib-Walter[3], Manuela Köpfli[1], Heidrun Kopp[9], Enrico van der Loo[10], Daniel Mata Flores[7], Diego Mercerat[7,11], Raphael Moser[2,5], Anthony Sladen[7], Fabian Walter[2,12], Diego Wasser[3], Ethan Welty[3], Selina Wetter[13] & Ethan F. Williams[1]

Interactions between melting ice and a warming ocean drive the present-day retreat of tidewater glaciers of Greenland[1-3], with consequences for both sea level rise[4] and the global climate system[5]. Controlling glacier frontal ablation, these ice–ocean interactions involve chains of small-scale processes that link glacier calving—the detachment of icebergs[6]—and submarine melt to the broader fjord dynamics[7,8]. However, understanding these processes remains limited, in large part due to the challenge of making targeted observations in hazardous environments near calving fronts with sufficient temporal and spatial resolution[9]. Here we show that iceberg calving can act as a submarine melt amplifier through excitation of transient internal waves. Our observations are based on front-proximal submarine fibre sensing of the iceberg calving process chain. In this chain, calving initiates with persistent ice fracturing that coalesces into iceberg detachment, which in turn excites local tsunamis, internal gravity waves and transient currents at the ice front before the icebergs eventually decay into fragments. Our observations show previously unknown pathways in which tidewater glaciers interact with a warming ocean and help close the ice front ablation budget, which current models struggle to do[10]. These insights provide new process-scale understanding pertinent to retreating tidewater glaciers around the globe.

Tidewater glacier calving and submarine melt, collectively called frontal ablation, are interconnected processes that are the main driver of mass loss from the Greenland Ice Sheet (GrIS)[3]. A simplified theory that relates calving and submarine melt describes a sequence of events that begins as warm Atlantic waters from subpolar origins travel north. These waters traverse the continental shelf and travel at depth into the fjords to reach the ice fronts of tidewater glaciers[8]. There, they become turbulently entrained in buoyant subglacial discharge plumes, cause submarine melting and thereby lead to undercut ice faces. This undercut geometry is mechanically unstable, and therefore enhances calving rates[11], which further drives accelerated mass loss at the scale of the entire GrIS[1].

Recent observations have challenged this simplified theory of ice–ocean interaction and found that plume models seem to underestimate submarine melt by up to two orders of magnitude[10], suggesting that other processes control submarine melt. The tightly packed broken ice blocks floating in fjords, called mélange[12], have also been shown to strongly suppress calving activity. Furthermore, the detailed processes involved in calving are still poorly constrained by direct observations and hence limit the related understanding and models. More specifically, potential feedbacks of calving events on fjord circulation, thermal structure, and hence oceanic melt remain hidden. Resolving these apparent gaps in our understanding of ice–ocean interaction is crucial for modelling frontal ablation and predicting the future of the GrIS. Obtaining the required process-scale data remains, however, challenging because of the hazardous conditions near the ice front.

Today, advances in submarine fibre sensing are transforming marine geophysics[13] but have not yet been used to study ice–ocean interactions. By repeatedly sending laser pulses into fibre-optic cables and measuring the backscattered light as the pulses travel through the fibre, Distributed Acoustic Sensing (DAS) and Distributed Temperature Sensing (DTS) convert optical fibres into linear arrays of thousands of seismo-acoustic and temperature sensors. The DAS technology is based on measuring phase changes in the Rayleigh-backscattered laser light, which can represent both variations in strain and the index of refraction in the fibre and may be caused by vibrations, pressure and

[1]Department of Earth and Space Sciences, University of Washington, Seattle, WA, USA. [2]Laboratory for Hydraulics, Hydrology and Glaciology (VAW), ETH Zurich, Zurich, Switzerland. [3]Department of Geography, University of Zurich, Zurich, Switzerland. [4]Department of Earth and Climate Sciences, Tufts University, Medford, MA, USA. [5]Swiss Federal Institute for Forest, Snow and Landscape Research WSL, bâtiment ALPOLE, Sion, Switzerland. [6]Extreme Environments Research Laboratory, École Polytechnique Fédérale de Lausanne, Lausanne, Switzerland. [7]Université Côte d'Azur, Observatoire de la Côte d'Azur, IRD, CNRS, Géoazur, France. [8]Department of Geophysics, Stanford University, Stanford, CA, USA. [9]GEOMAR Helmholtz Centre for Ocean Research Kiel, Kiel, Germany. [10]Department of Environmental System Science, ETH Zurich, Zurich, Switzerland. [11]CEREMA, Direction Territoriale Méditerranée, Nice, France. [12]Swiss Federal Institute for Forest, Snow and Landscape Research WSL, Birmensdorf, Switzerland. [13]Institut de Physique du Globe de Paris (IPGP), Université Paris Cité, Paris, France. ✉e-mail: graeffd@uw.edu

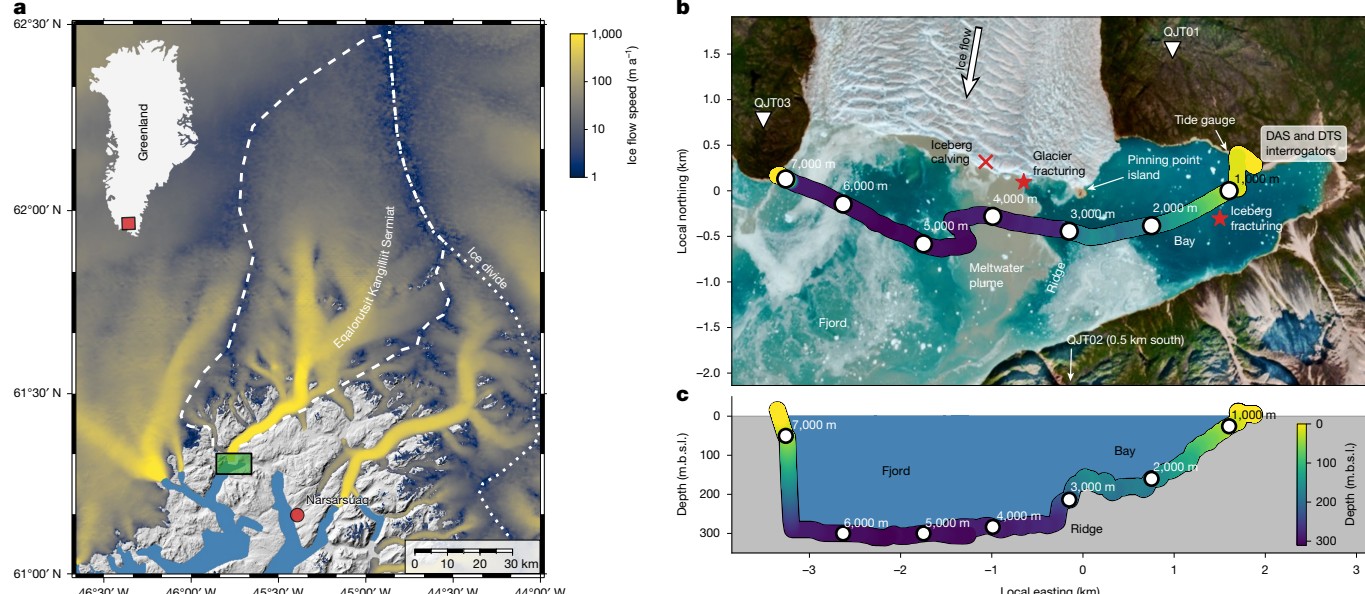

**Fig. 1 | Overview of seafloor fibre sensing at the calving front of EKaS. a**, Map of EKaS and location in Greenland (red rectangle) with colour-coded ice-flow speed and drainage basin outline (white dashed line)[51]. **b**, Magnification into the green rectangle in **a**. Layout of seafloor fibre-optic cable on satellite imagery (Sentinel 2 from 7 August 2023). The thick yellow–green–purple-shaded line shows the vessel track during cable deployment with bathymetry interpolated from the depth sonar of the vessel and labelled OD along the cable

as white circular markers. The specific locations of iceberg calving and both fracturing events presented in Fig. 3 are marked as red symbols. Seismometers QJT01 and QJT03 are located east and west of the calving front. QJT02 is about 0.5 km south of the map extent (Extended Data Fig. 2). **c**, East–West cross-section with bathymetry along the cable, including along-cable ODs. m.b.s.l., metres below sea level.

temperature fluctuations. By contrast, DTS measures the intensity of Raman-backscattered laser light from inelastic interactions between temperature-dependent molecular vibrations within the fibre, allowing us to determine strain-independent absolute temperature[14] (Methods). For both DAS and DTS, the travel time between laser light emission and the detection of the backscattered light determines the optical distance (OD) to the sensing locations along the fibre, which are discretized and referred to as 'channels'. Real-time data telemetry from these channels to the recording units, called 'interrogators', is established by the travelling laser pulses themselves. This allows for deployments in harsh cryospheric environments, whereas the interrogators can remain housed in more hospitable locations on land. Compared with traditional oceanographic and glaciological instrumentation, distributed fibre sensing has the benefit of millisecond temporal resolution, metre-scale spatial sampling and apertures of up to 150 km.

Here we demonstrate the value of distributed fibre sensing to investigate ice–ocean interactions along the calving front of the glaciers of Greenland at the example of Eqalorutsit Kangilliit Sermiat (EKaS; also known as Qajuuttap Sermia[15]), a large tidewater glacier in South Greenland (Fig. 1a). The drainage basin of EKaS is 5,800 km², making up around 0.3% of the GrIS area. Ice-flow velocities at the 3.5-km-wide terminus (grounded 280–300 below sea level; Fig. 1b,c) range between approximately 5 m day⁻¹ in August–September and 15 m day⁻¹ in May–June. This results in about 3 km³ yr⁻¹ ice discharge[16]. On 6 August 2023, we deployed a 10-km-long subsea fibre-optic cable on the seafloor across the fjord, at a few hundred metres distance from the calving front (Fig. 1b, Methods and Extended Data Fig. 1a,b). From 9 to 29 August 2023 (21 days), we used this cable for simultaneous DAS and DTS to monitor the seismo-acoustic wavefield and temperature along the fibre-optic cable continuously (Extended Data Figs. 1c and 2).

By sensing the seismo-acoustic and thermal wavefield along the calving front, our observations show a detailed coupled process chain surrounding iceberg calving and its interaction with the fjord (Fig. 2). First, small-scale fracturing generates seismic and acoustic waves associated with the shortest time scales in the calving process (<0.1 s

period). The iceberg detachment and impact on the water surface then progressively generate seismic waves that propagate along the seafloor–water interface (0.1–10 s period), tsunamis (5–30 s period) and finally disruptions to the fjord ocean waters in the form of propagating internal gravity waves (IGWs) (15–30 min period) and transient currents (>15 min period). After detachment, icebergs gradually decay by fracturing into broken fragments, building up the ice mélange in the fjord. Below, we explain and discuss these observations in more detail, highlighting their importance in the larger context of the retreating GrIS.

## Small-scale fracturing

Ice fracturing events (Fig. 2a) have timescales of only about 10–100 ms, but frame the entire glacier calving process chain. Related to the growth of 10–100 m scale fractures[17], fracturing events mark both the initiation of iceberg detachments (Fig. 3a) and the final disintegration of icebergs in the fjord (Fig. 3b). Small-scale glacier fracturing is pervasive during our entire observation period, with rates of several events per minute, indicating persistent forcing of the damage processes that will eventually culminate in iceberg detachment[18]. In this context, the observing capability of DAS is superior to other instrumentation: whereas recordings of terrestrial seismometers are dispersed and lack interpretable phase arrivals (Fig. 3a,b, top)—therefore impeding precise event location—the DAS record sections show crisp phase coherence of acoustic waves above 20 Hz (Fig. 3a,b, dotted black lines).

The seismic wavefield excited by small-scale fracturing events illuminates the seafloor landscape. Highly dispersive Scholte waves, that is, waves that propagate at the water–sediment interface[19], indicate a thick sedimentary cover that probably affects ice front stability and glacier dynamics[20]. Sharp wave reflections mark where a submarine ridge pierces these sediments as it extends southwards from the pinning point island of the ice front (Fig. 1b,c; 3,000 m OD in Fig. 3a). Wave conversions excite higher-order interface wave modes travelling at about 800 m s⁻¹, indicating sedimentary layering[21] (Fig. 3a, dashed black line). Water surface reflections of the acoustic wave are prominent 0.2 s after

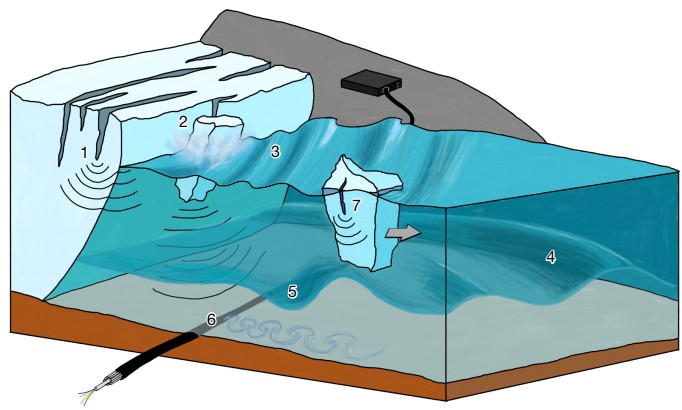

**Fig. 2 | Illustration of the calving process chain observed by fibre sensing.** Glacier fracturing is observed by fibre sensing through its acoustic signature travelling through the fjord waters (1). Fracturing events coalesce into iceberg detachments that emit interface waves propagating along the seafloor (2). These detachments excite calving-induced tsunamis at the water surface that cause pressure perturbations along the fibre-optic cable (3). Calving-induced IGWs in the stratified fjord waters cause temperature variations at a given depth (4). Calved-off icebergs drift away from the glacier terminus and drag internal wave wakes behind them, which cause seafloor cooling by vertically displacing the stratified fjord waters (5). The internal wave wakes cause transient enhanced seafloor currents that generate vibrations in the cable through vortex shedding (6). Finally, icebergs disintegrate by fracturing, again detected by fibre sensing through its acoustic signals (7).

the direct arrival (4,000–5,000 m OD in Fig. 3a), with some events exhibiting as many as four reflections (Extended Data Fig. 3). Composing a substantial part of the underwater soundscape, these impulsive acoustic waves propagate through the entire water column and therefore may be used as a tool for ocean thermal tomography across the ice front[22].

## Iceberg detachment

We observe the moment of iceberg detachment (Fig. 2b) occurring on timescales of seconds to minutes, depending on iceberg size and detachment mechanism. Together with submarine melt, iceberg detachment accounts for about half of the recent GrIS mass loss[23]. By detecting the latter, seismic recordings can unambiguously distinguish the two processes, in particular, for submarine calving events that are invisible to satellites. Our DAS observations excel in this regard: ice front detachments are recorded at a rate of 1–2 events per minute (Fig. 3c), typical durations of 10–20 s and bandlimited energy between 0.1 Hz and 10 Hz (Fig. 3d). At the low-frequency end, these signals correspond to the dynamics of large rotating iceberg detachments, and on the high frequency end to the timescale of air cavity collapse, as falling ice blocks plunge into the fjord waters[24,25].

Submarine seismo-acoustic observations can distinguish between subaerial and the 'hidden' submarine detachment events[26]. Therefore, they provide an essential observational constraint on the representation of calving in ice-flow models used for projecting the long-term evolution of glaciers and ice sheets. Our DAS records show that the seismo-acoustic wavefield produced by calving events and recorded at the seafloor predominantly comprises Scholte wave energy. The characteristic dispersion of these interface waves is evident by a shift to higher frequencies later in the wave train, becoming increasingly pronounced towards larger optical distances (5,000–6,000 m OD in Fig. 3d). Synthetic travel time curves determine the wave propagation speed to 230 ± 40 m s⁻¹ in the 1–10 Hz frequency band (Methods). This is much slower than the Scholte waves excited by fracturing events in about 1 km distance to the calving front and suggests a substantial increase in sedimentation rate towards the glacier terminus, as Scholte

waves are slower in freshly sedimented and unconsolidated interface material[21]. Faster (≥800 m s⁻¹; Fig. 3d, blue curve) 1–10 Hz arrivals near the noise floor are usually prevalent at the eastern fjord wall (OD 2,800–3,000 m) and in the shallow bay (OD 1,000–2,000 m) but are only weak directly behind the submarine ridge (OD 2,000–2,800 m). Characterized by an emergent, incoherent onset and a coda lasting about 10–15 s, we interpret these waves as higher-order Scholte waves, similar to those excited by small-scale fracturing and probably excited similarly by acoustic waves—here from the water cavity collapse—coupling into the sediments. These acoustic emissions are important in discriminating between submarine and subaerial iceberg detachments, as the former typically does not form a water cavity and therefore lacks these acoustic emissions and the cascading higher-order Scholte waves that are excited by them. In total, we detected about 56,000 iceberg detachments in the DAS data, of which around 35,000 had higher-order Scholte wave signals above the noise floor in the bay, therefore setting a upper limit on the submarine-to-subaerial calving event ratio of 2:3.

By sensing calving-induced Scholte waves, our seafloor DAS resolves and locates tens of thousands of iceberg detachments as small as about 100 m³. This is much below the detection threshold of most other methods, such as remote sensing platforms, local seismometer networks, time-lapse imagery and terrestrial interferometric radar (TRI) scans. More specifically, our DAS-based calving catalogue comprises 200 times more detections and a 40% larger cumulative detachment volume than the TRI scans detected (Methods and Extended Data Fig. 4). The only previous seafloor-seismic study near a glacier[27] relied on a single ocean bottom seismometer and, therefore, could not image the wavefield spatially resolved. This contrasts with our DAS records, which identify Scholte waves as the dominant carrier of the calving signal in the submarine environment. Scholte-to-surface converted waves appear to be a weak component of the seismic wave field observed by our terrestrial seismometers (Fig. 3d orange line and simultaneous arrival at seismometer at around 9 s)—a component that has not been identified in previous terrestrial seismic records at other calving fronts, to our knowledge. The strong sensitivity of seafloor DAS to Scholte waves enables sensing submarine calving, which is difficult to sense with terrestrial seismometers and other instrumentation[28]. Quantifying submarine calving volumes is of particular interest, as they may substantially contribute to the frontal ablation budget, which submarine melt parameterizations struggle to retrace.

## Calving-induced tsunamis

In the calving process chain, the detachment of large (≥50,000 m³) icebergs transfers gravitational potential energy to the water column through the excitation of tsunamis (Fig. 2c), that is, transient surface gravity waves (SGWs) in the water, which we observe with DAS[29]. These tsunamis have typical wave periods of multiple 10 s and can resonate in the fjord for more than 1 h (ref. 30). Similar to landslide-generated tsunamis in narrow fjords, these calving-generated tsunamis can pose substantial natural hazards to local populations[31], prompting calls for tsunami early warning systems (TEWS)[32]. Although DAS-based TEWS have shown promising far-field results[33], our study marks one of the first DAS-based near-field tsunami observations.

We observed several hundred tsunamis during our study period, with wave heights of up to 1.6 m (trough to crest) at a tide gauge in the bay (Fig. 4a; see Fig. 1b for location). The largest tsunami was excited by the largest calving event during our study period with a subaerial calving volume of about 300,000 m³, as determined by our TRI (Fig. 4d,e; see Extended Data Fig. 2 for location). The resulting tsunami wave was detected within our DAS array, about 700 m west of the pinning point island, and propagated with a group velocity of 31 m s⁻¹ at 30-s wave period before slowing down by 50% in shallower waters (Fig. 4a). At the subsea ridge, significant long-period tsunami wave energy was reflected into the fjord (Fig. 4a at 3,500 m OD and 150 s).

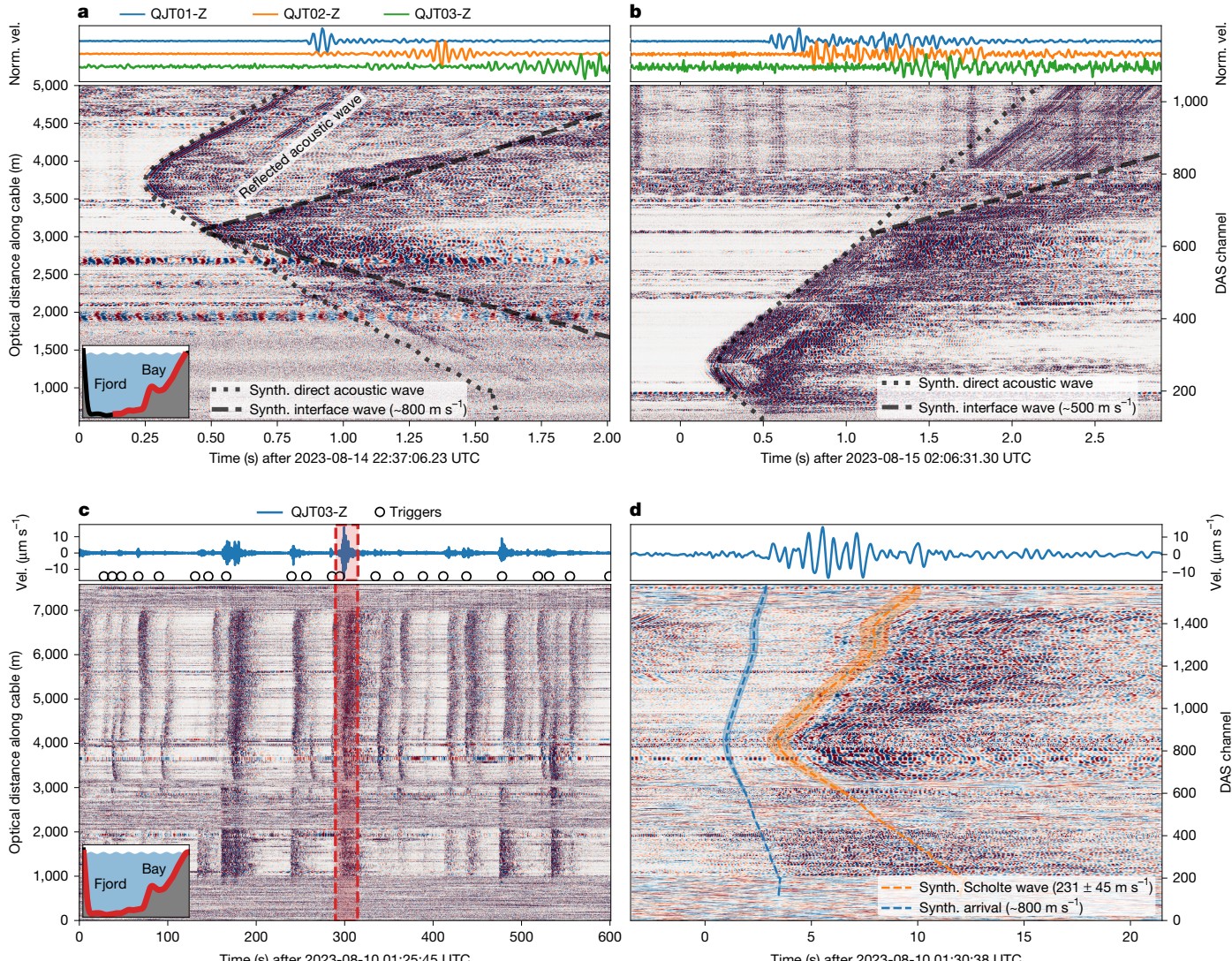

**Fig. 3 | DAS recording ice fracturing and iceberg calving. a**, Normalized DAS strain record (>20 Hz) of ice fracturing originating at the eastern part of the calving front (red cross in Fig. 1b). Blue indicates compressional, red extensional strain along the cable distance (left axis), and corresponding channels (right axis). The inset in the lower left corner shows an East–West cross-section of the fjord, resembling Fig. 1c, with cable sections used for this plot marked in red. Dotted and dashed lines indicate simulated arrival times. The dotted line indicates an acoustic wave travelling at 1,500 m s$^{-1}$. The dashed line indicates an interface wave travelling at about 800 m s$^{-1}$ from the sediment basin backscatter. **b**, Iceberg fracturing originating from the bay (red star in Fig. 1b), approximately above the cable segment at 1,400 m OD. The acoustic wave (simulated arrival as dotted line) is scattered stronger than in **a**. In the deep fjord sediments (OD > 3,000 m), a slower (about 500 m s$^{-1}$) wave propagates further along

the cable. We plot only ODs up to 5,000 m here, as this was the operational cable length at the time of the shown events. Corresponding normalized vertical component seismograms of broadband seismometers (see Fig. 1b for location) are shown at the top. **c**, A 10-min normalized DAS strain record filtered between 1 Hz and 10 Hz showing iceberg calving events (dark vertical bands). The top panels show the corresponding vertical seismograms (velocity) of the broadband seismometer station QJT03 (Fig. 1b), and circles indicate triggered calving events. **d**, Magnification into the 25-s-long record in the red marked region of **c**. Dashed lines (blue and orange) indicate modelled arrival times of both the Scholte wave and a faster propagating wave. At about 9 s, the Scholte-to-surface converted wave arrives at QJT03. Synth., synthetic; Norm., normalized; Vel., velocity.

These characteristics suggest that a fjord TEWS could provide about 2 min of warning time over distances of about 3 km. Facilitated by a unique ice-front parallel cable layout, tsunami epicentres are efficiently located in our dataset through the hyperbola vertices in the DAS wavefield recordings. Apart from this unique feature, the tsunami wave dispersion also makes it possible to determine the epicentral distance of fibre sections more distant from calving events. A frequency–wavenumber (f–k) transform (Fig. 4b) shows that the tsunami follows the SGW deep-water limit, in which the group velocity is dependent only on the frequency. Relating the rate of frequency shift d$f$/d$t$ to the event distance $D = (g/4\pi)(df/dt)$ (ref. 34) ($g$ is the acceleration due to gravity), we confirm an epicentral distance $D = 3 \pm 1$ km to the easternmost

DAS channels in the bay for the above-mentioned event, in agreement with the TRI-derived location (Fig. 4c). As a source-independent measurement, our observations demonstrate that existing subsea telecommunication fibre-optic cables close to Greenlandic villages may be used as DAS-based TEWS, also for tsunamis caused by nearby iceberg-capsizing[35].

## Internal gravity waves

We observe large iceberg detachments to also excite IGWs[36] (Fig. 2d) — that is, waves that oscillate within a fluid medium along density gradients. In the stratified fjord waters, we observe persistent but variable

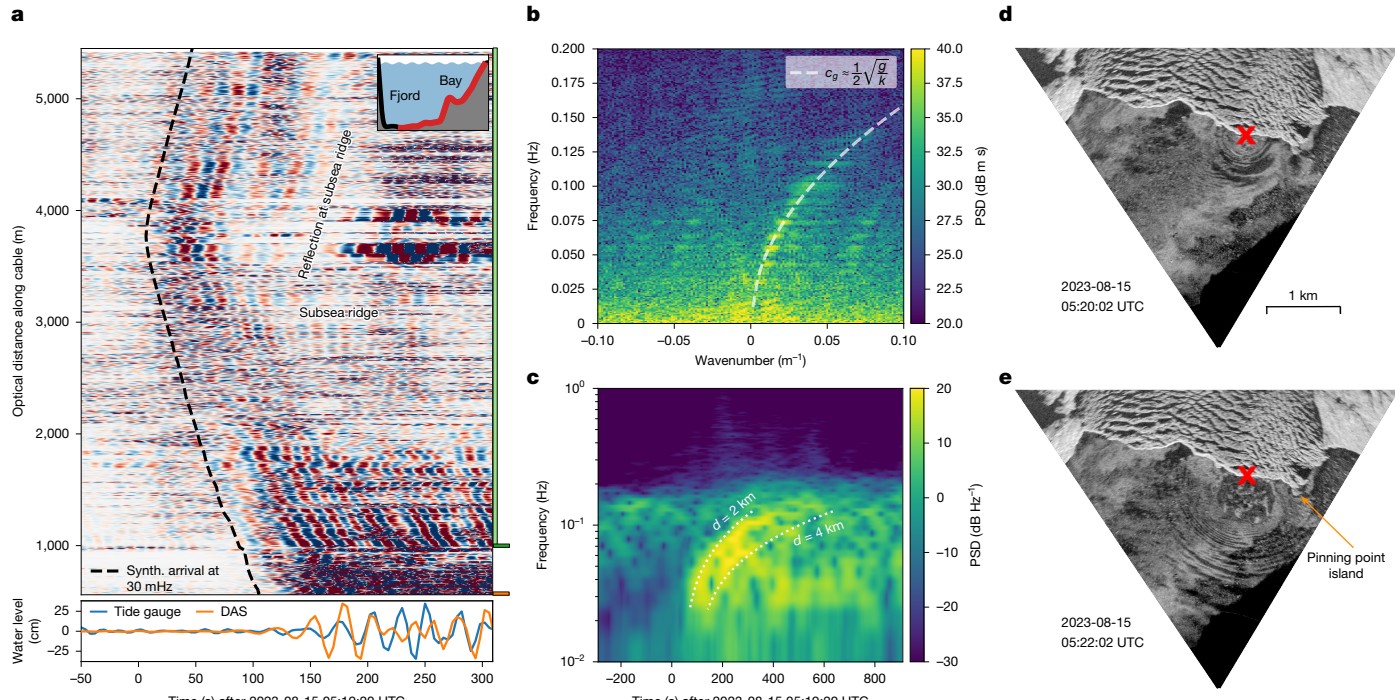

**Fig. 4 | DAS recording calving-induced tsunami. a**, Raw DAS record of a calving-induced tsunami (location in **d**,**e**) filtered between 30 mHz and 60 mHz and normalized per channel. The black-dashed curve is the modelled arrival time for an SGW with a frequency of 30 mHz. At the bottom of the plot, the tide gauge record at the eastern cable landing is shown alongside the mean DAS signal over the 50-m (10 channels) cable section closest to the tide gauge (downsampled to 0.25 Hz to match tide gauge sampling). The tide gauge is co-located with the lowest DAS channels in **a** (see orange marker on the right of **a**). **b**, Frequency–wavenumber (f–k) plot of the tsunami DAS recording from 1,000 m OD onwards (light green marker to the right of **a**). The dashed line is the theoretical dispersion relation in the deep-water limit (>30 m for relevant frequencies). **c**, Spectrogram of a 30-m stacked DAS section between 1,010 m and 1,040 m OD with a 2-min window size (dark green marker to the right of **a**). The dotted lines are the theoretical dispersion relation marking the expected tsunami wave arrival from 2 km and 4 km distance travel path with a constant 300 m water depth. **d**,**e**, Ortho-rectified TRI backscatter images of the calving-induced tsunami recorded 2 min apart. Each scan takes about 10 s. Calving event location marked with red cross (latitude, longitude: 61.3088, −45.7740), as also shown in Extended Data Fig. 2b. Synth., synthetic. Scale bar, 1 km (**d**,**e**).

IGW activity with oscillation periods varying between about 15 min and 30 min at various depths[37] (Fig. 5). IGWs induce elevated water velocities and, therefore, probably enhance submarine melt at the glacier terminus by convective heat transfer[38]. At our site, IGWs perturb a fjord rest state consisting of a thermohaline staircase (Fig. 5a,b). The staircase steps are a striking feature in our DTS temperature record, formed by abrupt steps of up to 0.5 °C over several metres. Conductivity–temperature–depth (CTD) casts confirm that these temperature steps co-locate with salinity and hence density step changes (Extended Data Fig. 5). Thermohaline staircases near glaciers[39] and icebergs[40] are formed by double-diffusive convection—temperature and salt diffuse at different rates—between the freshwater from subglacial discharge, submarine glacier melt and iceberg melt. This mechanism requires low current speeds and the absence of shear-driven mixing[41]. Turner angles, as a measure of the stability regime of the water column, are calculated as $T_u = -50° ± 6°$ below the glacially modified waters at about 0–100 m depth, that is, subglacial discharge mixed with oceanic water (Methods and Extended Data Figs. 6a and 7) confirm that diffusive convection is active[42]. The resulting staircase pattern with sharp temperature steps allows us to monitor IGW activity with DTS and DAS.

The amplitude of the IGW field is highly variable over time, with occasional impulsive wave packets (for example, 28 August 4:00 and 18:00 in Fig. 5a), which are evident in both the DTS and DAS data at the thermohaline staircase steps (Fig. 5 and Methods). Travel times suggest IGW propagation speeds of the order of 1 m s⁻¹, whereas oscillation periods just below the buoyancy frequency imply nearly horizontal wave propagation (Methods and Extended Data Fig. 6b). Furthermore, transient small-scale temperature inversions are visible,

in which warmer water temporarily is located above colder water (for example, 28 August 0:00 around 2,950 m OD and Fig. 5b), possibly formed by subglacial discharge, meltwater intrusions or breaking internal waves. These thermal features may be caused by overturnings that would result in a regime change from diffusive convection to salt fingering, in which parcels of warmer saltier water sink into colder and fresher water[42] (Methods). Observing these processes helps us to understand the temporal variability of the vertical mixing processes, which, in turn, alters the transfer of heat to the ice front and influences submarine melt by disturbing the boundary layer between the glacier ice and the fjord waters.

Through our continuous DAS and DTS observations, we can directly link transient IGW packets (Fig. 6a,b) as originating from calving events of ≥50,000 m³ subaerial volume[36] (measured by TRI; Extended Data Fig. 8) and thereby estimate their influence on submarine melt. The measured vertical isothermal displacements of the IGW of up to about 60 m (trough to crest) correspond to a wave-induced temperature change of up to 1.2 °C (Extended Data Fig. 9a). Estimated vertical water velocities associated with the IGWs reach up to around 5 cm s⁻¹ (Methods and Extended Data Fig. 9b), probably constituting a dominant component of the velocity field at the terminus away from the subglacial discharge plume. Velocities of the order of cm s⁻¹ are between the purely buoyancy-driven regime, in which submarine melt is independent of velocity, and the shear-driven regime, in which melt scales linearly with velocity[43]. Assuming submarine melt to scale linearly with near-ice water velocity[44], we calculate submarine melt rates of up to 1 cm per IGW wave cycle (Methods and Extended Data Fig. 9c). As we observe repeated large calving events to continuously keep

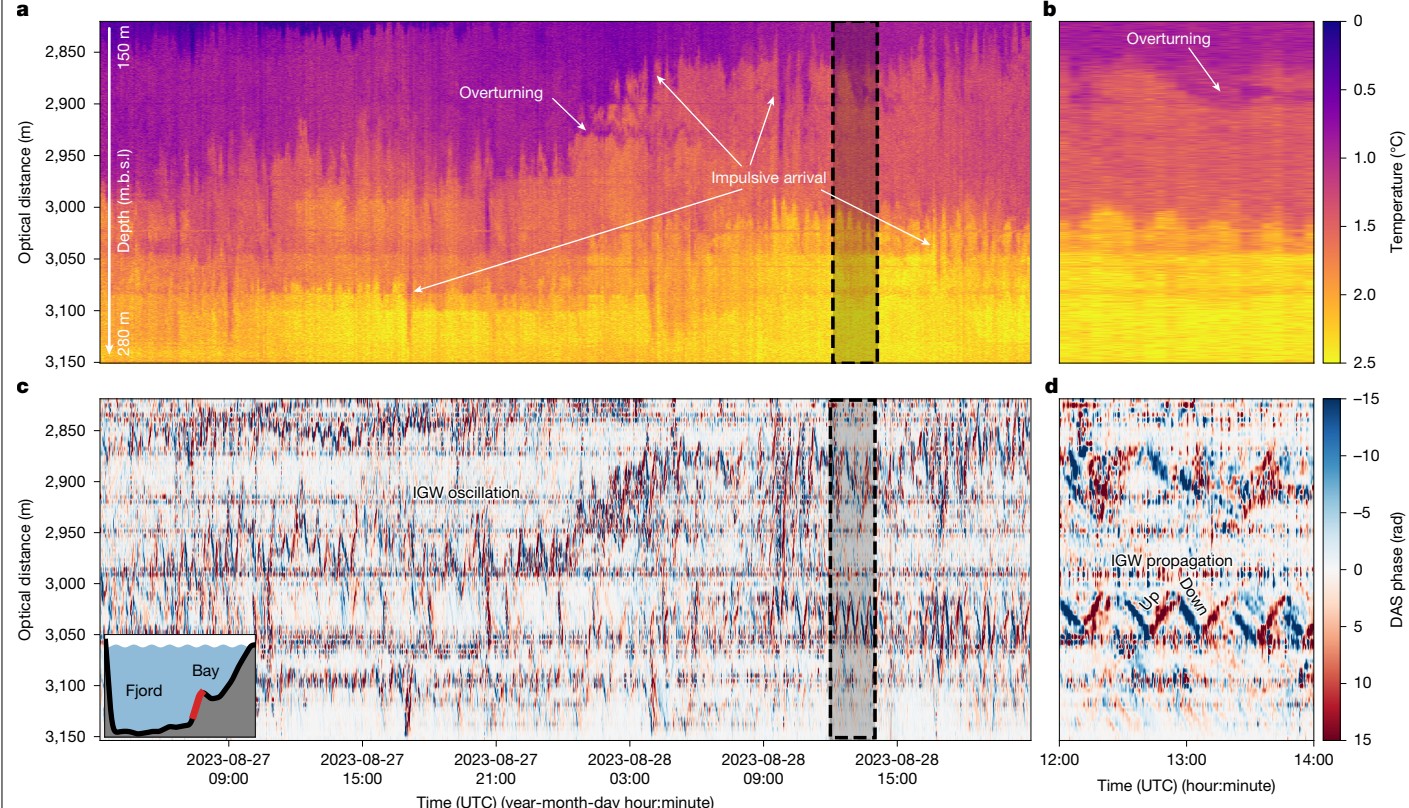

**Fig. 5 | Continuous internal wave activity vertically displacing the temperature layers of the fjord. a**, A 1.5-day-long temperature profile along the sloping cable from DTS (see inset at the bottom left corner of **c** and depth scale on the left of **a**). IGWs with a 15–30 min period are predominantly visible where the vertical temperature gradient is the largest (sharp transitions between yellow–orange, orange–purple and purple–blue). **b**, Magnification into the marked region of **a**. **c**, Unfiltered DAS raw phase record of the same cable section as in **a**. For the IGW frequencies shown (<10 mHz), the measured phase is proportional to the strain rate along the cable, caused by temperature change. **d**, Magnification into the marked region of **c**, showing the vertical displacement of the sharp temperature steps by changes in strain rate propagating up and down. Positive (red) values indicate an increase in temperature and negative (blue) values a decrease. m.b.s.l., metres below sea level.

the IGW activity excited (Fig. 5)—in contrast to a previous study that assigned the IGW activity to the subglacial discharge plume[38]—daily IGW-induced melt rates may reach 1 m day$^{-1}$. This is comparable to the mean ice-flow speed across the calving front during our observations (around 4.5 m day$^{-1}$), and ultimately suggests that, in a low background current environment, calving-induced IGW activity may cause a significant fraction of ambient submarine melt away from the subglacial discharge plume—a process that has not been appreciated so far.

## Seafloor cooling and currents

Following detachment, newly formed icebergs gain momentum and drift away from the calving front. Time-lapse imagery indicates that this happens with speeds up to several m s$^{-1}$, with iceberg drafts probably exceeding 100 m depth. As icebergs pass over the cable, we observe IGW wakes (Fig. 2e) that manifest as transient seafloor cooling in DTS records (Fig. 6c). Isotherms at this point are first heaved up by the wake directly behind the iceberg and then surge down below their resting level[45]. During the heaving phase, the seafloor temperature stays constant because the vertical temperature gradient in between the thermohaline steps and, in particular, below the deepest step is small (Extended Data Fig. 5). During the isothermal downward motion, colder waters from above reach the seafloor and result in a temperature drop along the cable. During DTS-measured temperature drops of up to 0.8 °C (Fig. 6c), DAS phase records typically saturate but show hyperbolic arrivals from a propagating internal wave

front (Fig. 6d and Methods). These observations again highlight the unique observing power of our optical fibre platform. Whereas CTD casts rarely resolve internal wave activity and moored acoustic Doppler current profilers lack spatial coverage during large iceberg passages, seafloor DAS and DTS resolve these features spatio-temporally. The recordings, therefore, show a mechanism—that is, the IGW wake excitation—that reduces the momentum of the iceberg during its drift, alters the thermal structure of the fjord and—as a source of kinetic energy originating at the glacier terminus—probably increases the ice front ablation.

Synchronized with seafloor cooling events, we observe enhanced cable vibrations from elevated seafloor current speeds (Fig. 2f) ranging between 5 and 20 cm s$^{-1}$ (Fig. 6e,f). Through vortex shedding—that is, the repeated formation and release of eddies behind an object—on the leeside of the fibre-optic cable, seafloor currents generate harmonic strain oscillations that are coherent over 20–50 m. These occur in cable sections probably suspended over bathymetric depression or in loose contact with the sediments[46]. The strain signal measured over these sections is steadily about 10 times higher than for the rest of the cable[47] (Fig. 6e). During iceberg transits over the cable, the vortex-shedding frequency linearly increases with seafloor current speed, ranging up to 2–10 Hz with overtones exceeding 50 Hz. From this, narrow-banded tension-dominated natural frequencies of the cable get excited, which scale inversely with the length of the suspended cable section. The resulting spectral signature allows for measurements of current speed perpendicular to the fibre-optic cable and perpendicular to the calving front (Fig. 6f). These transient

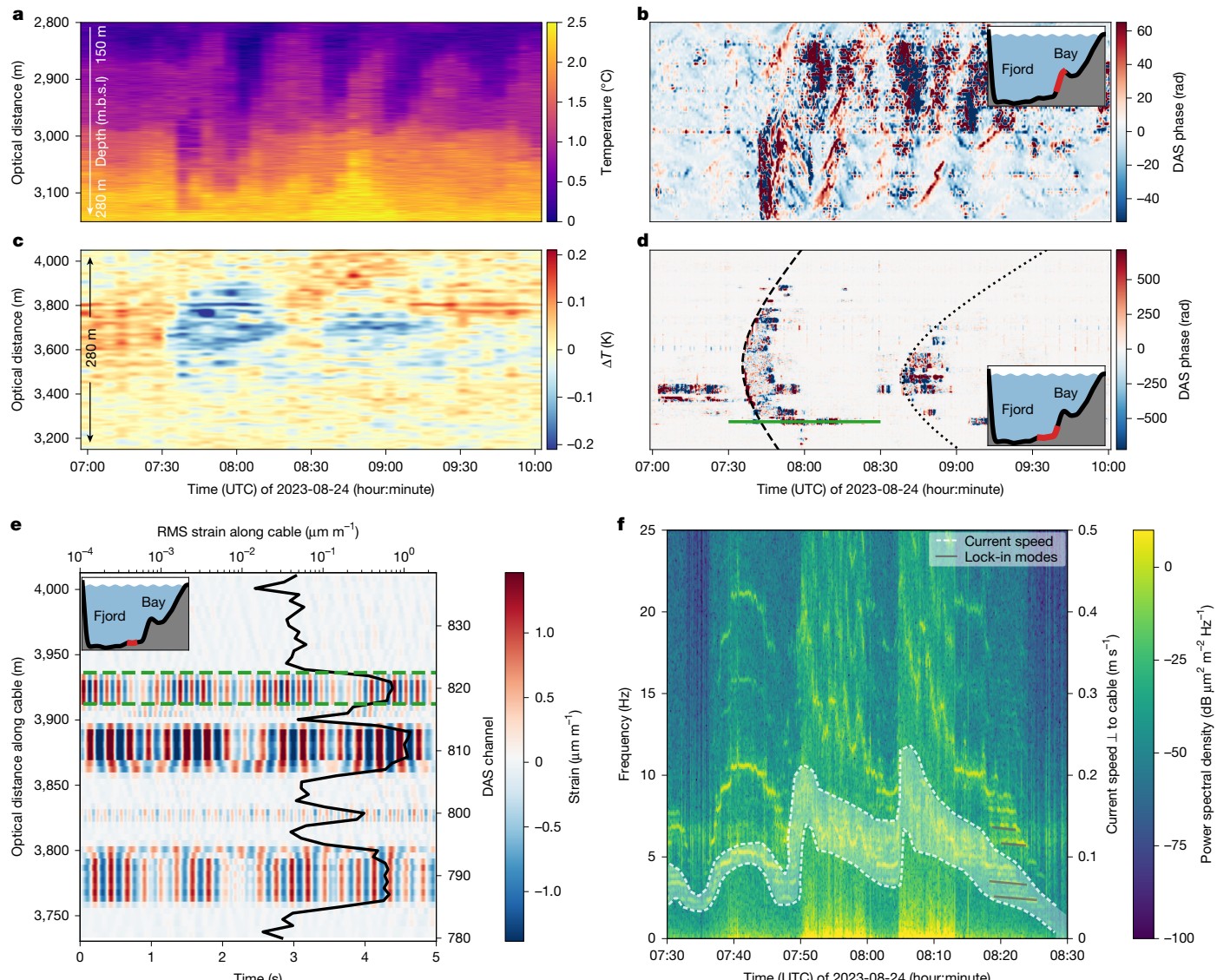

**Fig. 6 | Transient internal waves and seafloor currents excited by iceberg calving and successive iceberg drift. a**, DTS record of a transient IGW train at the eastern fjord wall over 3 h caused by an iceberg calving event. **b**, Corresponding DAS record, which saturates for the strongest IGW signals. **c**, DTS deep-seafloor temperature anomaly relative to the median temperature during the passage of two IGW wave wakes (compare with **d**) caused by a passing iceberg (confirmed from time-lapse imagery). **d**, Corresponding DAS record with dashed and dotted hyperbolas emphasizing the moveout of the internal wave wakes. The green line marks channel 820 from 7:30 to 8:30 UTC, which is used for **f**. **e**, DAS record of a 300-m-long deep-seafloor cable section with coherent harmonic oscillations of a few Hz. **f**, A spectrogram of channel 820 (within the section marked in green in **e**) shows vortex-induced vibrations with varying frequencies in the fibre-optic cable from which we calculate current speeds perpendicular to the cable. m.b.s.l., metres below sea level.

seafloor currents below drifting icebergs may alter the heat transport to the ice front and influence the overall submarine ablation budget.

## The fate of icebergs

The last process that we observe in the calving process chain is the disintegration of icebergs (Fig. 2g). As for glacier fracturing at the beginning of the calving process chain, the acoustic wavefield of impulsive (about 10–100 ms), high-frequency (>20 Hz) fracturing events originating from icebergs in the fjord tracks the decay of icebergs after detachment from the glacier. Submarine DAS allows for a clear distinction between events happening at the ice front versus iceberg fracture and subsequent disintegration (locations shown in Fig. 1b; Methods), which our field team repeatedly could witness to be preceded by gunshot-like bangs from the iceberg. Figure 3b shows a particular iceberg fracturing event from the shallow bay. This iceberg was probably grounded at the time of fracturing, as indicated by the substantial degree of direct seafloor coupling apparent in the higher-order Scholte waves, compared with free-floating iceberg fracturing (Extended Data Fig. 3). When the direct acoustic wave (dotted black line in Fig. 3b) travels over the ridge and down into the deep fjord sediments, it again excites a higher-order Scholte wave, but with slower velocity (around 500 m s$^{-1}$) than for the ice front fracturing event (approximately 800 m s$^{-1}$, Fig. 3a). A broadband seismometer placed on bedrock west of the fjord (QJT03; see Fig. 1b for location) registers the event before the acoustic wave travels across the fjord, suggesting that the first arrival must have travelled through the faster ice or bedrock with about 4,500 m s$^{-1}$. This complicated propagation path probably explains the jumbled nature of these events in the broadband seismic data, impeding precise event location, as opposed to the clear phase arrivals seen in DAS recordings. Through iceberg fracturing, subsea DAS directly observes ice mélange formation, which

has been shown to buttress the ice front and suppress iceberg calving when sufficiently compact[48].

## Calving–fjord interactions

Our results show that, using a single subsea cable, we observe the glacier calving process chain and quantify several previously undocumented components therein. More specifically, we were able to characterize sediment cover and locate glacier pinning points on the seafloor landscape; track the evolution of thermohaline staircases; monitor calving activity; observe calving-induced tsunamis, IGWs and seafloor currents; track glacier and iceberg fracturing; and explain the details of submarine seismo-acoustic wave propagation (Fig. 2). All of these observations establish pieces that help to understand tidewater glacier ice–ocean interactions and close the submarine frontal ablation budget. Although we focused on a single study site, we expect that similar processes occur at tidewater glacier termini across the globe.

The power of fibre sensing for studying ice–ocean interactions rests on (1) its high spatial sampling density and wide spatial coverage and (2) its ability to offer ice front-proximal and ice front-parallel observing geometries. We expect that submarine fibre sensing will be applied to diverse studies in other fjord systems with calving glaciers in Greenland and the Arctic, as well as coastal Antarctica.

Whereas the study of submarine melt and calving can be motivated by the desire to improve projections of global-scale sea level rise, the study of SGWs also provides means to address the local-scale hazard posed by fjord tsunamis. Our study demonstrates the ability of submarine DAS to survey calving-induced fjord tsunamis. We suggest that this observational ability extends to landslide- and iceberg capsizing-induced fjord tsunamis, which may originate closer to coastal settlements[31,32,49] and generalize that DAS-based TEWS could offer widespread utility in the near future.

Our results also indicate a potential, previously unappreciated calving multiplier, through which calving activity creates waves and currents that enhance submarine melt that, in turn, favours additional calving[50]. We assume this process to be operative at other tidewater glaciers across the world and suggest it to be most effective in areas with low current speeds, that is, where calving causes perturbations to the water velocity field that are large compared with the background ocean water velocity. This multiplier effect is, therefore, expected to have the largest relative importance outside the melt season, when strong subglacial discharge plumes are lacking, and submarine melt in traditional theories and models would be suppressed. This result potentially explains why certain calving front models underestimate submarine melt by up to two orders of magnitude[10] and should be investigated further. And although it is long known that glacier frontal ablation consists of both front melt and calving, our work suggests that these two processes are more closely interconnected than previously thought.

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

# Methods

## Study site

Eqalorutsit Kangilliit Sermiat (EKaS; also known as Qajuuttap Sermia) is, to our knowledge, the only major Greenlandic tidewater outlet glacier that has continuously gained mass over the past three decades[52]. At the calving front, the Eqalorutsit Kangilliit fjord is 280–300 m deep and filled with sediments, resulting in a flat bathymetry across the fjord. At the eastern part of the calving front, a pinning point bedrock island with about 100 m in diameter emerges from a subsea ridge that separates the deep fjord from a shallower bay (Fig. 1b). Calving at EKaS mainly consists of smaller collapses with infrequent rotational events as large as about $10^6$ $m^3$. Between November and June, the fjord is usually covered with dense ice mélange near the calving front. Subglacial discharge is in summer of the order of 200 $m^3$ $s^{-1}$ and is at times visible through the upwelling of turbid meltwater plumes at the calving front. Episodic outburst floods from subglacial and marginal lakes originate at the confluences of EKaS with its northwestern tributary glaciers.

## Fibre-optic cable deployment

The deployed fibre-optic cable is a 10-km-long Universal Cable from the supplier NBG. It has a diameter of 4 mm and is armoured by 12 steel wire stands. Two single-mode (SM) and two multi-mode (MM) fibres are guided in a gel-filled steel tube. The cable has a mass of 40 g $m^{-1}$ (3,200 kg $m^{-3}$ density) and was spooled onto a wooden cable drum with a total weight of 500 kg. We deployed the cable by hand from the stern of the ice-strengthened vessel Adolf Jensen (Extended Data Fig. 1a,b) and synchronized the cable output with the travelled GNSS track distance. Cable sections in shallow water near the eastern shore were deployed from a small, motorized landing boat. The 250-m-long cable section on land leading to the interrogators was trenched in about 20 cm depth. By contrast, the entire 7-km-long submarine cable section was deployed without trenching, by letting the cable sink to the seafloor.

A strong optical loss in both SM fibres at approximately 960 m OD along the cable occurred already during the cable deployment (Extended Data Fig. 2b), forcing us to operate our DAS measurements atypically on one of the two MM fibres. These only experienced a co-located loss of 3 dB. During our 3-week-long fibre-optic measurements, the cable suffered multiple damages. From 10 August 05:41 UTC and 17 August 04:40 UTC onwards, we were not able to measure any backscattered light from farther than 5,435 m and 4,135 m OD, respectively, which effectively shortened our cable to terminate at the centre of the calving front eventually. From 10 August 12:10 UTC onwards, a substantial 3.5 dB loss in the MM fibre was present at 681 m OD. We attribute these losses, on one hand, to kinks in the fibre-optic cable formed by seafloor currents dragging the cable and tightening loops that formed from too much slack. On the other hand, we attribute the losses to bends in the fibre because of icebergs becoming stranded on the cable in the shallow bay waters during low tide.

## DAS interrogator setup

We deployed a Sintela ONYX v.1.0 interrogator for DAS at the northern shore of the bay east of the calving front (Fig. 1b). The interrogator was situated in a waterproof aluminium box together with 12× Yuasa 65 Ah 12 V gel batteries powered by 4 × 300 Wp Goal Zero Ranger solar panels (Extended Data Fig. 1c). We continuously recorded strain along one of the two available MM fibres in our cable at 2 kHz sampling rate, 4.79 m gauge length and identical channel spacing between 9 and 29 August (1,587 initial seismo-acoustic sensing sections). A data gap occurred between 20 August 13:55 UTC and 21 August 13:00 UTC (23 h) because of an unexpected laser shutdown of the interrogator that was detected only a day later during maintenance. While recording, we additionally saved directly downsampled data at 100 Hz, 10 Hz and 1 Hz sampling rates, creating a dataset of 13 TB in total. Each time the effective cable length diminished, we adjusted the interrogator settings to record only the functional cable length.

## DTS interrogator setup

Co-located with the DAS interrogator, we deployed a Silixa XT-DTS M interrogator for DTS. Similar to the DAS setup, the interrogator was situated in a waterproof aluminium box with 6× Yuasa 65 Ah 12 V batteries powered by 1 × 300 W Goal Zero Ranger and multiple smaller solar panels (33–68 W) mounted around the box for stand-alone operation (Extended Data Fig. 1c). We operated the DTS in single-ended configuration on one of the two MM fibres with a channel spacing of 25 cm and an integration time of 5 min (34,448 initial sensing sections). As a first temperature reference, we spooled up 25 m of cable inside a styrofoam box that was placed inside the larger aluminium box on top of the batteries. As a second temperature reference, we spooled up 25 m of cable inside another insulated box and buried it at 50 cm depth in the soil. Both temperature reference sections were equipped with a PT100 thermometer directly connected to the DTS interrogator. Moreover, we used an RBRduet[3] temperature and pressure sensor at the eastern cable landing location to measure near-surface water temperature. On the west shore, we spooled up the remaining cable and put it into an insulated aluminium box together with a Geoprecision M-Log 5 W temperature logger. As for the DAS setup, we adjusted the DTS interrogator settings each time the effective cable length diminished.

## Additional instrumentation

In July 2022, we installed three Nanometrics Trillium Compact broadband seismometers with a flat response between 120 s and 100 Hz on the hilltops east, south and west (QJT01, QJT02 and QJT03) of the calving front of EKaS and sampled them at 200 Hz. Owing to fox bite damage, the sensors were only running continuously from 2 August 2023 12:25 UTC (QJT03), 11 August 2023 17:20 UTC (QJT01) and 14 August 2023 17:20 UTC (QJT02). On 5 August, we measured 22 vertical CTD (conductivity, temperature and depth) profiles with an RBRconcerto[3] profiler from the R/V Adolf Jensen along the fibre-optic cable deployment route parallel to the calving front. A weather station (Decagon) co-located with the interrogators collected air temperature, solar irradiation, precipitation and relative humidity in 10-min intervals starting from summer 2022. For the first days of our fibre-optic measurements until 15 August, a terrestrial radar interferometer (Gamma Portable Radar Interferometer, TRI) situated on the hill south of the calving front was scanning the calving front of EKaS in 1-min intervals and monitoring ice-flow speed and calving activity (timing, volume and locations)[53]. A time-lapse camera co-located with the radar took images of the calving front in 2-min intervals until 15 August, and in 20-min intervals subsequently. Where the fibre-optic cable enters the bay, a combined temperature and pressure sensor (RBRduet[3]) recorded tides, wave heights and surface water temperature at a 3-s interval.

## DTS calibration

We use the Python package dtscalibration (ref. 54) for calibration of the raw Raman Stokes and anti-Stokes backscattering recordings from the Silixa XT-DTS M. We corrected for the optical losses in the fibre that occurred during the deployment, by applying a correction factor to fibre sections after the loss such that the corrected Stokes and anti-Stokes intensity in a 30-m region around the optical loss is linear. The 3.5-dB loss that occurred on 10 August 12:10 UTC altered the recorded backscattered light for multiple tens of metres nonlinearly. We corrected for this particular optical loss by multiplying all measurements and fibre sections before and after the loss by the ratio of the last measurement before and the first measurement after the loss occurred, assuming only small fluctuations in temperature over the 5-min measurement interval. This procedure results in Stokes and anti-Stokes data, as would be expected without optical losses along the fibre.

After this preprocessing, we followed a single-ended calibration procedure in ref. 54 to derive the differential loss $\Delta\alpha$ along the fibre, the sensitivity of the Stokes and anti-Stokes scattering to temperature $\gamma$, which depends on the fibre material, and a lumped factor $C$ accounting for sensitivity of the detector, the number of molecules involved in scattering and the wavelength dependency backscattered light. The temperature for each fibre section is then obtained with the measured Stokes $P_S$ and anti-Stokes $P_{AS}$ intensity by

$$T(x, t) \approx \frac{\gamma}{\ln\left(\frac{P_S}{P_{AS}}\right) + C(t) + \Delta\alpha x} \quad (1)$$

We solve for $\gamma$ and $\Delta\alpha$ by fitting the DTS temperatures at reference sections (1) inside an insulated box within the aluminium box that houses the interrogator; (2) inside an insulated box 50 cm deep in the soil; and (3) a cable coil in shallow water at the shore with co-located temperature probes. Furthermore, we require the cable sections close to the shore east and west of the fjord to match while the full fibre-optic cable length is operational. We then calibrate the temperature for the entire 3-week-long measurement with fixed $\gamma$ and $\Delta\alpha$ values. Residuals of the calibration are shown in Extended Data Fig. 11.

## DAS phase to strain conversion
We convert raw phase $\Delta\Phi$ (units of radians) recordings of the Sintela ONYX interrogator to units of strain using

$$\Delta\epsilon = \frac{\lambda}{4\pi n_{\text{eff}} \, x_g \psi} \Delta\Phi \quad (2)$$

with the vacuum wavelength of the laser $\lambda = 1{,}550.12$ nm, the effective refractive index of the fibre $n_{\text{eff}} = 1.4682$, the gauge length $x_g = 4.79$ m, and the strain-optic coefficient $\psi = 0.78$. The internal data flow of the interrogator applies a leaky frequency filter at 0.1 Hz. For frequencies $\lesssim 10$ mHz, this filter acts as a differentiator, which results in phase measurements proportional to strain rate, instead of strain. Therefore, we show the DAS data in units of strain, raw phase record or normalized per channel, depending on the frequency of interest.

## DAS temperature response
In the DAS recordings, a temperature increase at a given segment of the fibre-optic cable will become evident as a positive strain-rate signal (red in Figs. 5c,d and 6b,d). This signal has two origins: (1) the thermal expansion of the fibre-optic cable with increasing temperature; and (2) the increase of the refractive index with temperature will be sensed by a positive phase change:

$$\Delta\phi = (\alpha_T + \xi)\Delta T \times \frac{4\pi n_{\text{eff}} \, x_g}{\lambda} \quad (3)$$

with the thermal expansion coefficient $\alpha_T = 5 \times 10^{-7}$ K$^{-1}$, the thermo-optic coefficient $\xi = 6.8 \times 10^{-6}$ K$^{-1}$ (ref. 55). As for frequencies $\lesssim 10$ mHz, our DAS recordings are proportional to strain rate (see previous section), our interrogator's response is then about $\Delta\phi = \frac{1}{13 \times 10^{-3}(\text{K s}^{-1})} \times \frac{dT}{dt}(\text{K s}^{-1})$, meaning that a temperature change by 13 mK s$^{-1}$ will result in a 1 rad signal. $\Delta\phi \gtrsim 100$ over less than a minute caused by the seafloor temperature drops associated with internal wave wakes resulted in faulty phase unwrapping and, therefore, subsequently wrong phase measurements[56] (Fig. 6b,d). For the maximum seafloor temperature drop of 0.8 K over less than 100 s (interrogator response is proportional to strain), the expected phase change would be $\Delta\phi \approx 330$.

## Cable location
We expect the cable to sink to the seafloor in about 15 min. The cable sinking speed is approximately determined from the force balance between the drag force per unit length

$$\frac{F_D}{l} = \frac{1}{2}\rho_{\text{water}} v^2 c_d \, d \quad (4)$$

acting on the cable during sinking with the density of water $\rho_{\text{water}}$, the sinking velocity $v$, the drag coefficient $c_d = 1.0$, the cable diameter $d = 4$ mm and the gravitational force on the cable

$$\frac{F_D}{l} = -g\left(m_{\text{cable}} - \rho_{\text{water}}\pi\frac{d^2}{4}\right) \quad (5)$$

with the gravitational acceleration $g$, and the mass density per unit cable length $m_{\text{cable}} = 0.04$ kg m$^{-1}$, resulting in a sinking speed of $v \approx 0.4$ m s$^{-1}$. Currents in the fjord are expected to be strongest at the water surface and up to 1 m s$^{-1}$ (ref. 57). We, therefore, expect the cable to not be farther off the vessel deployment track than a few hundred metres in the deep fjord sections and virtually underneath the vessel track to within a couple of ten metres in the bay in which strong surface currents are absent.

To further constrain the location of individual cable sections in the deep fjord (Extended Data Fig. 2a), we pick the Scholte wave arrivals from a calving event at 10 August 01:30:45 UTC (Fig. 3d) with a short-term-average/long-term-average (STA/LTA) trigger of 1 s/10 s in the data band-passed filtered between 1 Hz and 10 Hz (Extended Data Fig. 10a). The TRI located the origin of the tsunami caused by this event at 61.310, −45.783 (latitude, longitude; Extended Data Fig. 10c). From the picked arrival times and the event location, we then run a Bayesian inversion using a Markov-chain Monte-Carlo algorithm for locating individual cable sections in the deep fjord section of our fibre-optic cable installation[58]. Therefore, we follow the steps below:

1. We pick the calving front location at 40 points from Sentinel 2 imagery and interpolate linearly between the picks to get a continuous one-dimensional function of the calving front location (longitude as a function of latitude).
2. Interpolate the cable deployment track west of the pinning point island, resulting in a coordinate for every DAS channel and consecutively resample this curve to only 11 anchor points—one at the west shore, one where the cable enters the deep fjord and nine equally spaced (OD) in between. These nine anchor points $(x, y)$ will be what we will invert for later $(2 \times 9 = 18$ parameters). We leave the boundary anchor points to be fixed, as we know the location at the west shore, and we assume that the cable sits directly below the deployment track in the bay.
3. Our observations are the relative arrival times of the Scholte wave at the cable, which we automatically picked from the DAS data (Extended Data Fig. 10a).
4. Our model ($M$) calculates the two-dimensional travel time between the calving event location and the DAS channel coordinates for a given velocity.
5. We define a prior for our Bayesian inversion, assigning an a priori probability to the parameters that we will invert for and which are the 9 anchor points with $x$ and $y$ coordinates each $(x_n, y_n)$, a Scholte wave velocity $v$, the East–West coordinate of the calving event location d$x$ (North–South coordinate is derived from the shape of the calving front in (1)), and an absolute time offset d$t$ for when calving event happened $p(x_n, y_n, v, dx, dt)$. We assign a zero probability to the set of inverted parameters if the Euclidean distance between the anchor points is larger than the OD along the cable. We further assign a Gaussian probability distribution to the anchor point locations with a width of 1,000 m and centred around the initial position from the cable deployment track. Last, we allow for small shifts of the calving event location and assign it a Gaussian weighting with a width of 100 m around the radar-derived calving location.
6. For calculating the likelihood of the chosen parameters (anchor points, Scholte wave velocity, calving event location and event time), we do a cubic interpolation of the anchor points, resulting in a location for each channel. We then run the arrival time model

for the chosen parameters and define the likelihood as the absolute difference between the modelled and the picked arrival times $p(t_k|x, y, x_n, y_n, v, dx, dt)$, with arrival times $t_k$ and the interpolated locations for each DAS channel $x, y$.

7. We calculate the logarithmic posterior probability as the sum of the logarithmic prior and the logarithmic likelihood

$$p(x_n, y_n, v, dx, dt|t_k) \propto p(x_n, y_n, v, dx, dt) \, p(t_k|x, y, x_n, y_n, v, dx, dt). \quad (6)$$

8. We now initialize a Markov-chain Monte-Carlo sampler with 64 walkers for each inversion parameter, randomly distributed around the initial cable location using the affine invariant Markov-chain Monte-Carlo (MCMC) ensemble sampler emcee[57] for maximizing the logarithmic probability function from (6). We run the MCMC for 50,000 steps while checking for convergence. We select only the last 10,000 steps, resulting in a posterior probability distribution estimate of the parameters we inverted for−in particular, for our nine free anchor points.

9. We again do a cubic spline interpolation between the nine free and two constrained anchor points to retrieve a cable geometry for each accepted inversion. From this cable geometry set, we calculate the median and the standard deviation, corresponding to the most likely cable layout given the calving event picks as well as the Scholte wave speed (Extended Data Fig. 10a,b).

An accurate assignment of absolute depths to cable sections leading from the submarine ridge between the fjord and the bay down into the deep fjord was unsuccessful. An attempt to match DTS recordings on cable sections between 2,820 m and 3,150 m OD from 9 to 10 August to the CTD temperature profiles close to the ridge from 6 August resulted in an unphysical cable layout with depth differences between adjacent DTS channels larger than the channel spacing of 25 cm. The reason for this discrepancy is probably the daily temperature variability of up to 0.5 K for a given depth as well as the long-term variability of several 100 mK day$^{-1}$ that impedes a direct comparison of the DTS and the non-simultaneously recorded CTD profiles (Extended Data Fig. 5).

## Locating fracturing and detachment events

Our cable layout along the calving front spans an approximately linear array of sensors. For such a linear array, hyperbolic-shaped first wave arrivals are expected from point sources (see section 7.4.2 of ref. 59). In the case of a perfectly linear array, the point source origin is ambiguous and can be located on a plane that is perpendicular to the linear array. Our imperfectly linear cable layout resolves this ambiguity, allowing for event location in three dimensions. For locating both iceberg detachment and fracturing events, we fit synthetic travel time curves to the wave arrivals of the recorded DAS signal (Fig. 3a,b,d). In the case of acoustic arrivals from glacier and iceberg fracturing events, we assume a constant three-dimensional wave speed of 1,500 m s$^{-1}$ in water, and locate the event origin for the shown events at the ice front, the bay and the fjord (Figs. 1b and 3a,b and Extended Data Fig. 2a). In the case of Scholte wave arrivals from iceberg detachments, we assume a two-dimensional propagation along the water−sediment interface with an unknown constant wave speed. We then invert simultaneously for the event location and the wave speed, identifying the wavefield origin at the calving front (Fig. 3d and Extended Data Fig. 2a).

## Detachment event sensitivity estimate

The interferometrically derived subaerial calving volume integrated over 10 min from the TRI does not fully resolve the single calving events observed with DAS, thus hampering an unambiguous assignment of calving volumes to DAS observations. As a result, a TRI-derived calving-volume catalogue comprises 133 events in 10 days, compared to about 30,000 events observed with DAS over a similar 10-day time span (observational times do not fully overlap). Calving is expected to

represent a self-organized criticality, in which the calving event rate $R$ follows a power-law relationship with calving event volume $R \propto V^{-b}$ (refs. 18,60) and, therefore, allows us to estimate a detection threshold for our subsea DAS. An exponent $b = 0.79 \pm 0.06$ fits our TRI-derived calving volume number statistics, indicating that large calving events dominate mass loss (exponent <1). Extrapolating the cumulative volume number statistics from the TRI catalogue to the 200-fold event rate, we derive the calving event detection threshold of our DAS recordings as about 100 m$^3$, much below the TRI threshold of about 5,000 m$^3$ (Extended Data Fig. 4). The expected total calving volume observed with DAS then calculates to be around 40% larger than the total volume observed with TRI, because of the TRI missing small events. Comparing the ice flow based on a satellite-derived mean ice-flow velocity across the calving front of 4.5 m day$^{-1}$, a front width of approximately 3,500 m and an average height of about 80 m, and accounting for changes in the front position, the TRI detects only around 35% of the actual calved-off ice volume. However, owing to smoothing, 50% of the measured volume may get lost in the TRI detection process, increasing the TRI sensitivity up to about 70% of the actual calved-off volume. Now, considering the estimated cumulative calving volume detected with DAS is 40% larger than with the TRI, this means that DAS may detect up to about 98% of all solid frontal ablation. Note that, unlike the TRI, DAS is also able to detect submarine calving events.

## Calving location from tsunami dispersion

Calving-induced tsunamis propagate as linear SGWs. We apply a frequency−wavenumber ($f$−$k$) transform to the DAS data (Fig. 4b) and find good agreement between the observed $f$−$k$ energy distribution and the SGW dispersion relation

$$\omega(k) = \sqrt{gk \tanh(kh)} \quad (7)$$

with angular frequency $\omega$, wavenumber $k$, gravitational acceleration $g$ and water depth $h$ (ref. 61). We dominantly observe eastward tsunami propagation (that is, energy with positive wavenumber). The tsunamis mainly occur in the SGW deep-water limit ($h > 0.5\lambda$), where the group velocity

$$c_g \approx \frac{1}{2}\sqrt{\frac{g}{k}} \quad (8)$$

is independent of water depth. In the 300-m-deep fjord, this relation holds for wave numbers $k > 0.01$ m$^{-1}$.

Calving event spectrograms (Fig. 4c) show the arrival of low-frequency energy (about 30 s period) approximately 2 min earlier than higher-frequency energy (around 10 s period). By relating the rate of frequency shift to the distance $D$ of the wavefield origin[62],

$$D = \frac{g}{4\pi}\frac{df}{dt} \quad (9)$$

and fitting it to the spectrogram, we locate the calving event that caused the recorded tsunami 750 m west of the pinning point island, which agrees with TRI backscatter imagery (latitude, longitude: 61.3088, −45.7740; Fig. 4d,e). From this location, we simulate the tsunami arrivals in Fig. 4a using the Python package pykonal[63] based on the bathymetry and the general formula for the group velocity at 30 mHz:

$$c_g = \frac{1}{2}\sqrt{\frac{g}{h}\tanh(kh)}\left(1 + \frac{2kh}{\sinh(2kh)}\right) \quad (10)$$

## Fjord stratification stability analysis

Cold and fresh water overlying warm and salty water, as we observe, are preconditions for convective salt and heat transporting processes. The Turner angle

$$T_u = \arctan\left(\frac{R_\rho + 1}{R_\rho - 1}\right) \tag{11}$$

is a measure of the strength of the convection process[42,64].

$$R_\rho = \alpha\left(\frac{\partial T}{\partial z}\right) \Big/ \beta\left(\frac{\partial S}{\partial z}\right) \tag{12}$$

is the density ratio with the thermal expansion coefficient $\alpha$, the haline contraction coefficient $\beta$ and the vertical temperature and salinity gradients $\frac{\partial T}{\partial z}$ and $\frac{\partial S}{\partial z}$. Turner angles of $T_u = 50 \pm 6°$ (Extended Data Fig. 6a), calculated from our CTD casts, show that weak diffusive convection is present, particularly below the subglacial discharge intrusion at about 100 m depth, explaining the staircase pattern that we observe in the CTD casts (Extended Data Fig. 5).

## IGW analysis
We observe a typical triangular pattern with reversing polarity in the DAS records as propagating IGWs impinge on the steeply sloping seafloor cable (Fig. 5d). IGW frequencies of about 0.5–1.0 mHz (15–30 min period) are close to and slightly below the buoyancy frequency $N$, calculated from CTD casts (Extended Data Fig. 6b), representing the free oscillation mode of a stratified fluid (see equation 2.243 of ref. 65),

$$N = \sqrt{-(g/\rho_0)\frac{\partial\rho(z)}{\partial z}} \tag{13}$$

with reference fluid density $\rho_0$ and density gradient $\partial\rho(z)/\partial z$ with depth $z$. The mean buoyancy frequency below 150 m depth calculated from the CTD profiles is $N = 1.4 \pm 0.4$ mHz with a peak of $N = 2.3$ mHz at 193 m depth, at which a thermohaline step is located. With DAS and DTS, we predominantly measure the vertical displacement of the thermohaline steps. Along with the simplified IGW dispersion relation

$$\omega = N\cos(\theta) \tag{14}$$

observed IGW frequencies $\omega < 0.5N$ at the thermohaline steps imply $\theta > 60°$ with the vertical, and thus we can determine predominantly horizontal wave propagation. During the passage of an internal wave, the temperature change $\Delta T(z)$ at a given depth $z$ is assumed to be adiabatic, with variations only due to heaving of isotherms (vertical advection),

$$\Delta T(z) = w(z)\frac{\partial T(z)}{\partial z} \tag{15}$$

with vertical displacement $w(z)$. For our thermohaline staircase, we observe salinity to scale linearly with temperature, and to dominate the water density (Extended Data Fig. 7b), implying that the IGW signal measured with DTS and DAS will be strongest at temperature steps.

## DAS sensitivity to internal waves
The 5-min sampling of our DTS just barely samples the IGWs with a 15–30 min periodicity. Vastly improved resolution is offered by our DAS data with its 2 kHz sampling rate. For frequencies lower than 10 mHz, as is the case for the internal waves, our DAS phase recordings are proportional to strain rate (see previous sections). IGWs become evident by the vertical displacement of the temperature steps (Fig. 5b), as well as the horizontal propagation of IGWs along the cable (Fig. 5d). An upward heaving of the thermohaline staircase (increasing temperature at fixed depth) will be recorded as a positive strain rate, a downward motion as a negative strain rate[55]. The DAS record of IGWs, therefore, allows an independent measurement of the thermocline evolution that does not suffer from an absolute temperature calibration as the DTS

record requires. As DAS accurately resolves the IGW frequency, which we observe to be close to the buoyancy frequency, temporal variations in the density gradient can be resolved.

## Calving-induced submarine melt
We calculate the submarine melt rate at the calving front due to turbulent heat transfer from calving-induced IGWs following the commonly used three-equation model[44]. The first equation relates the freezing temperature in the boundary layer $T_b$ along the ice–ocean interface to salinity $S_b$ and pressure $p_b$

$$T_b = aS_b + b + cp_b \tag{16}$$

with empirical coefficients $a = -5.73 \times 10^{-2}$ °C psu$^{-1}$, $b = 9.39 \times 10^{-2}$ °C and $c = -7.53 \times 10^{-8}$ °C Pa$^{-1}$ (ref. 44). The second equation is the heat balance at the interface

$$\dot{m}\rho_i[c_i(T_b - T_i) + L] = \rho_w c_w\sqrt{C_D}\Gamma_T\,|u|\,(T - T_b) \tag{17}$$

with the melt rate $\dot{m}$, the density of ice $\rho_i$ and water $\rho_w$, the specific heat capacities of ice $c_i$ and water $c_w$, the drag coefficient $C_D = 0.01$, the turbulent transfer coefficient for heat $\Gamma_T = 0.01$, the absolute ice-parallel speed outside of the boundary layer $|u|$, the ice temperature $T_i$, and the water temperature outside of the boundary layer $T$ (measured by DTS). The third equation is the salt balance at the interface

$$\dot{m}\rho_i S_b = \rho_w\sqrt{C_D}\Gamma_S\,|u|\,(S - S_b) \tag{18}$$

with the turbulent transfer coefficient for salt $\Gamma_S = 3 \times 10^{-4}$, and the salinity outside the boundary layer $S$ (ref. 66).

We solve the three equations for the melt rate by using the linear relationship between the temperature and salinity from the CTD casts $S = 0.5$ psu °C$^{-1} \times T + 32.6$ psu (Extended Data Fig. 7b) and assuming that the ice temperature equals the boundary layer temperature $T_i = T_b$, and meaning that all available heat is used for melting, which introduces only a minor uncertainty. From the DTS data, we calculate vertical water velocities at the fibre-optic cable and assume these to be identical adjacent to the ice. With these, we calculate melt rates of up to 1.3 cm during a 3-h-long IGW wave train (0.07 mm min$^{-1}$) and peak values of up to 0.2 mm min$^{-1}$ (Extended Data Fig. 9). For IGW with amplitudes comparable to the water depth, we expect the horizontal flow component to contribute equally to the melt rate. Appreciating the effects of surface roughness, turbulence induced by released air bubbles from melted glacier ice, as well as the uncertainty in the turbulent heat transfer, and underestimations in the velocity field derived from the DTS data due to the finite temperature resolution, we expect melt rates of more than 1 mm min$^{-1}$ (refs. 67–69).

## Calculating seafloor currents from vortex-induced vibrations
During the passage of IGW wakes[70], we observe harmonic oscillations of the cable typically coherent over 20–50 m and strain signals about 10 times higher than for the rest of the cable. Two different types of vibrational modes with a distinct spectral signature are present. The first type constitutes tension-dominated cable modes with closely spaced frequencies (grey lines in Fig. 6f) following a linear sequence, with the $n$th harmonic frequency following

$$f_n = \frac{n}{2L}\sqrt{\frac{T}{m}}\,, \quad n = 1, 2, 3, \ldots \tag{19}$$

with the suspended cable length $L$, the horizontal tension in the cable $T$ and the cable mass per unit length $m = 0.04$ kg m$^{-1}$ (refs. 46,71). These first-kind cable modes are associated with lock-in vortex-induced vibrations (VIVs), establishing the natural frequencies of the suspended cable section.

The second mode type is characterized by widely spaced frequencies, also following a linear sequence $f_k \propto k$. They result from multiple lock-in harmonics $f_n$, excited sequentially as the lock-out VIV frequencies

$$f_k = \frac{k\,\upsilon_\perp \mathrm{St}}{d}\,,\ k = 1, 2, 3, \ldots \tag{20}$$

pass each of the lock-in frequencies $f_n$. Here $\upsilon_\perp$ is the ocean current speed normal to the cable axis, $\mathrm{St} \approx 0.2$, the Strouhal number describing the flow regime, and the cable diameter $d = 4$ mm. From the lock-out frequencies $f_k$, we determine the ocean current speed perpendicular to the cable (along the fjord) (Fig. 6e,f). During IGW wake passage, the fundamental mode lock-out VIVs ($k = 1$) at the flanks of the wake range between about 2 Hz and 12 Hz (region between dashed white lines in Fig. 6f), which translates to seafloor current speed between 0.05 m s$^{-1}$ and 0.2 m s$^{-1}$. Lock-in frequencies, in turn, vary only slightly over time (fundamental mode: about 2.5 Hz in Fig. 6f), as with increasing current, the cable tension $T$ increases, on which the lock-in frequency $f_n$ is only weakly dependent $f_n \propto \sqrt{T}$ (ref. 72).

## Data availability

Data and Jupyter Notebooks for reproducing the analysis of this study are available at Zenodo[73] (https://doi.org/10.5281/zenodo.15353304). Topography, ice-flow velocities and ice extent shown in Fig. 1a were provided by the Greenland Ice Sheet Mapping Project (GrIMP) through the National Snow and Ice Data Center (NSIDC). The background image of Fig. 1b and Extended Data Fig. 2a was provided by Copernicus Sentinel 2.

## Code availability

The code used in this study is available at Zenodo[73] (https://doi.org/10.5281/zenodo.15353304).

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

**Acknowledgements** We thank the people of Greenland for hosting us in their country. We thank everybody who contributed to the 2022/23/24 GreenFjord Cryosphere Cluster field campaigns, especially the 2023 crew of 60° N from Qaqortoq on the R/V Adolf Jensen. Broadband seismometers were provided by the Geophysikalischer Instrumentenpool (grant no. GIPP202206) of the GFZ German Research Center for Geosciences (GFZ). D.G., B.P.L. and R.J. were financed through the US NSF (grant nos. 2338502 and 2338503). This project received funding from the Swiss Polar Institute (project no. SPI-FLAG-2021-002). The solar-power stations for the DAS and DTS interrogators and the CTD were financed through the Einrichtungskredit from Andreas Vieli at the University of Zurich. The FiberLab of the University of Washington funded the fibre-optic cable through the Murdock Charitable Trust. J.S. holds the Ingvar Kamprad Chair for Extreme Environments Research sponsored by Ferring Pharmaceuticals. We also thank A. Feierabend, R. Häubi and R. Mardens for their outreach activities.

**Author contributions** D.G. designed the study together with A.V., B.P.L. and F.W.; D.G. led the fibre-optic cable deployment and interrogator and seismometer installation, and performed the data analysis. D.G. and B.P.L. formulated the manuscript, to which D.F., A.V., R.J. and J.S. contributed substantially. A.V. organized the GreenFjord Cryosphere cluster field campaigns. B.P.L., R.J. and D.G. acquired the funding for the fibre-optic sensing and salaries from the US National Science Foundation. J.S. and A.V. acquired the GreenFjord funding from the Swiss Polar Institute. B.P.L. assisted with the fibre-optic field deployment. A. Dachauer and A.K.-W. operated the terrestrial radar interferometer in the field and analysed its data. Financially supported by D.F., R.M. assembled the solar-power stations. E.F.W. and E.B. helped with internal wave data analysis. R.J., J.-P.A., D.M., A.S. and D.M.F. assisted with data interpretation. F.W. supported the broadband seismometer logistics. M.K., S.W., A. Dannowski, H.K., E.v.d.L., J.S., D.W., A.K.-W. and E.W. supported the fieldwork and were involved in fibre-optic cable deployment and the collection of various additional datasets (for example, CTD, time-lapse imagery, TRI and seismometers). All authors commented on the paper. From the eighth position onwards, co-authors are listed in alphabetical order.

**Competing interests** The authors declare no competing interests.

**Additional information**
**Correspondence and requests for materials** should be addressed to Dominik Gräff.

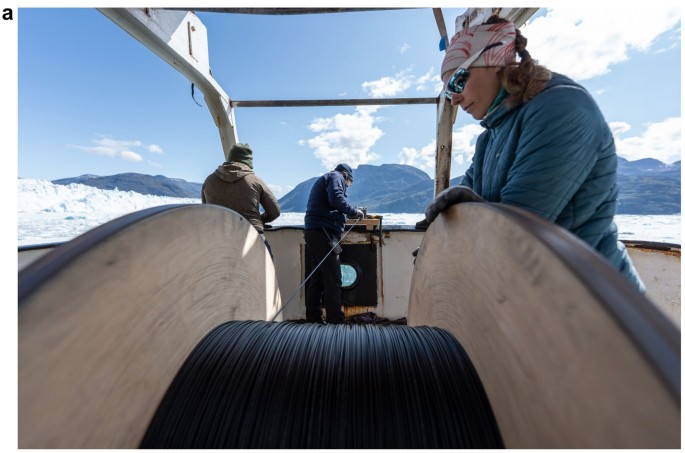

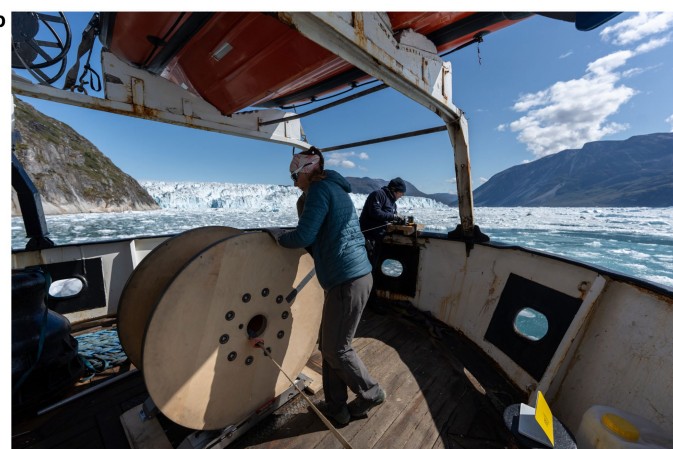

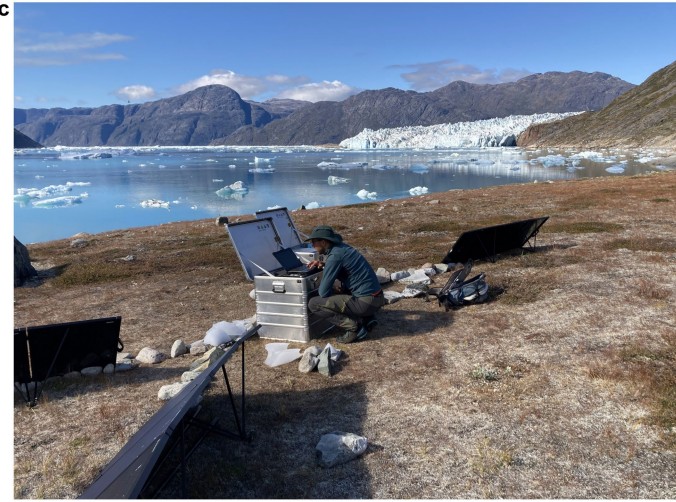

**Extended Data Fig. 1 | Pictures of the fibre-optic cable deployment and interrogator setup. a/b,** Pictures showing the manual deployment of the subsea fibre-optic cable from the ice ship R/V Adolf Jensen. **a,** View over the cable drum towards the cable counter and the stern of the vessel. **b,** Wooden cable drum was hand-propelled and controlled by one person. A second person supervised the cable length counting and synchronization of the deployment speed with the vessel speed (Credit: Richard Mardens). **c,** Picture showing the two (open) aluminum boxes that host the DAS and DTS interrogators and the solar panels (black) that power them. In the background, the eastern part of the calving front of Eqalorutsit Kangilliit Sermiat between the pinning point island and the eastern shore is visible (Credit: Manuela Köpfli).

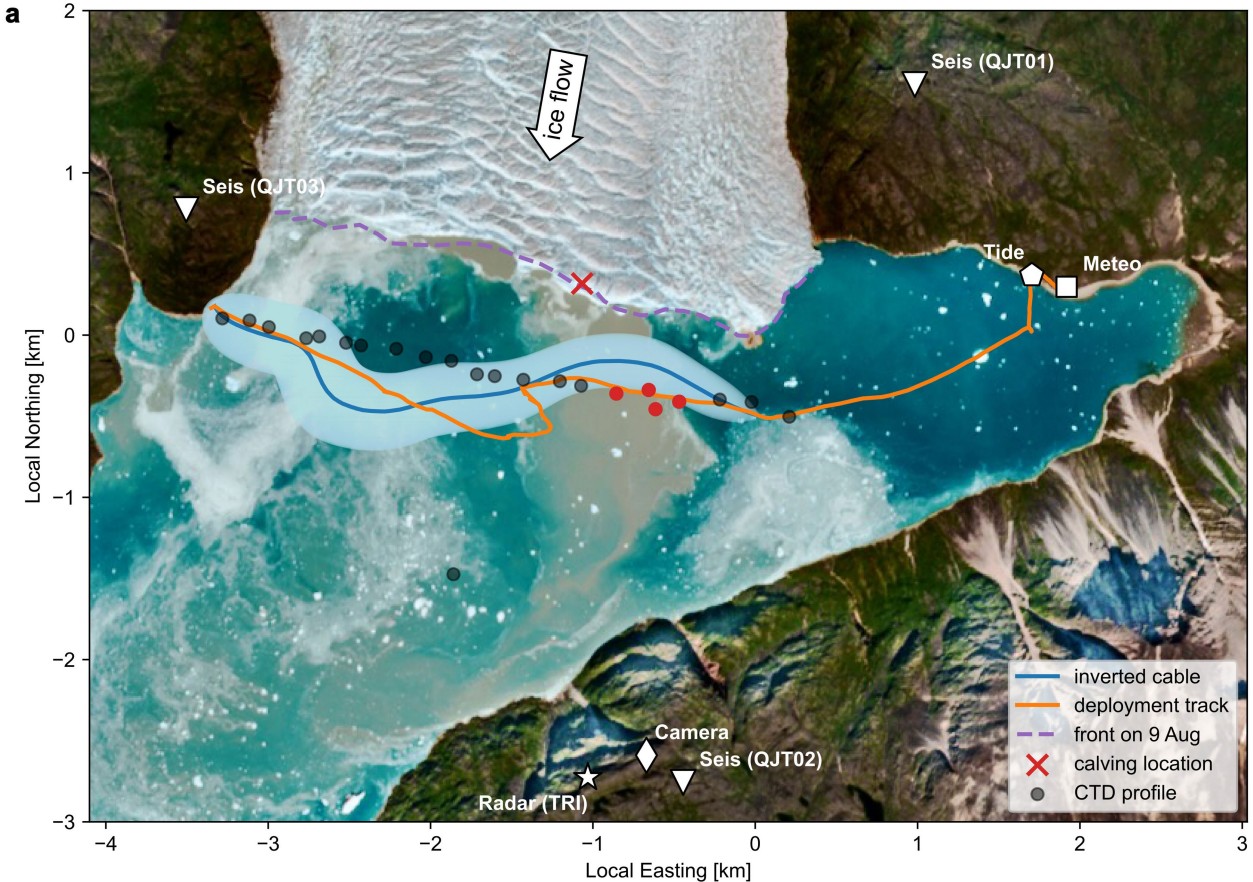

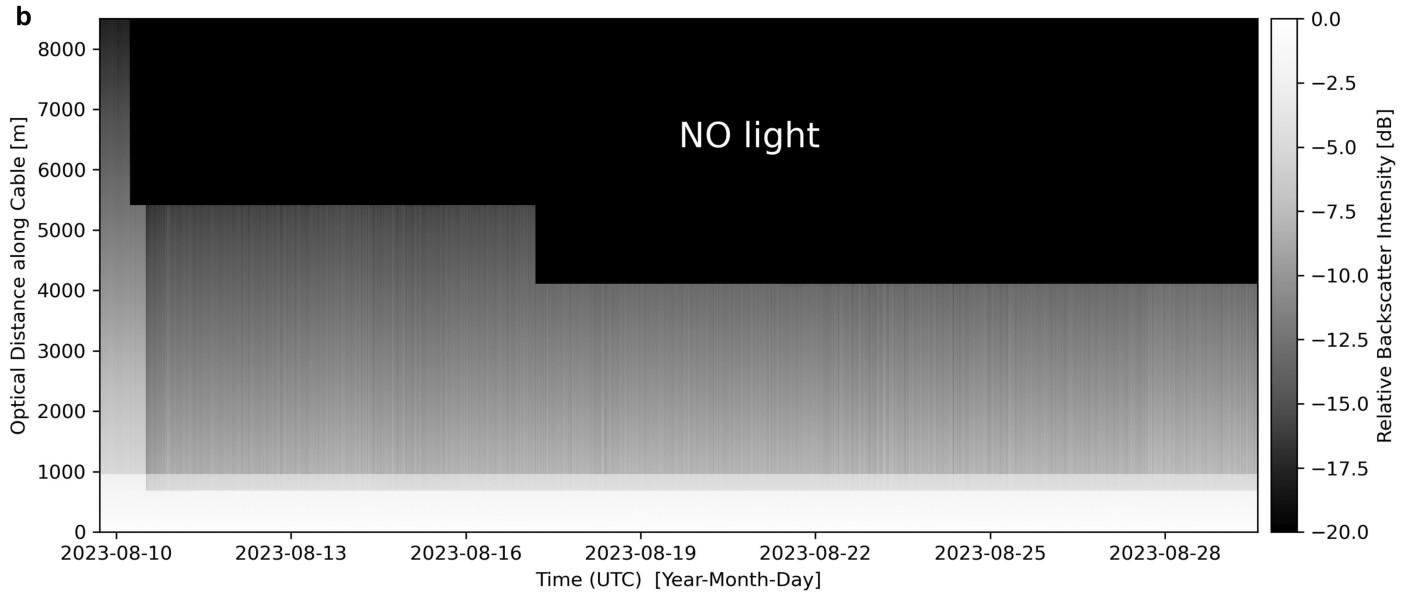

**Extended Data Fig. 2 | Cable layout in deep fjord, additional instrumentation, and operational cable length over time. a**, Orthophoto (Sentinel 2, 7 August 2023) with the cable deployment track and the expected cable layout (blue line with light-blue shading indicating 1σ uncertainties) in the deep fjord from a Bayesian inversion using a calving event location and Scholte wave arrivals recorded with DAS (see Extended Data Fig. 10 and Methods). White markers indicate location of seismometers (triangles), tide gauge (pentagon), meteorological station (square), time lapse camera (diamond), and terrestrial radar interferometer (star). Gray dots indicate CTD profiles. Red dots indicate CTD profiles included in Extended Data Fig. 5. **b**, OD along the fibre over time showing the sensed backscatter intensity (Raman-Stokes) on a logarithmic scale. Black colour means that no light (<20 dB of inserted intensity) could be transmitted to the corresponding OD and therefore no backscattered light was sensed.

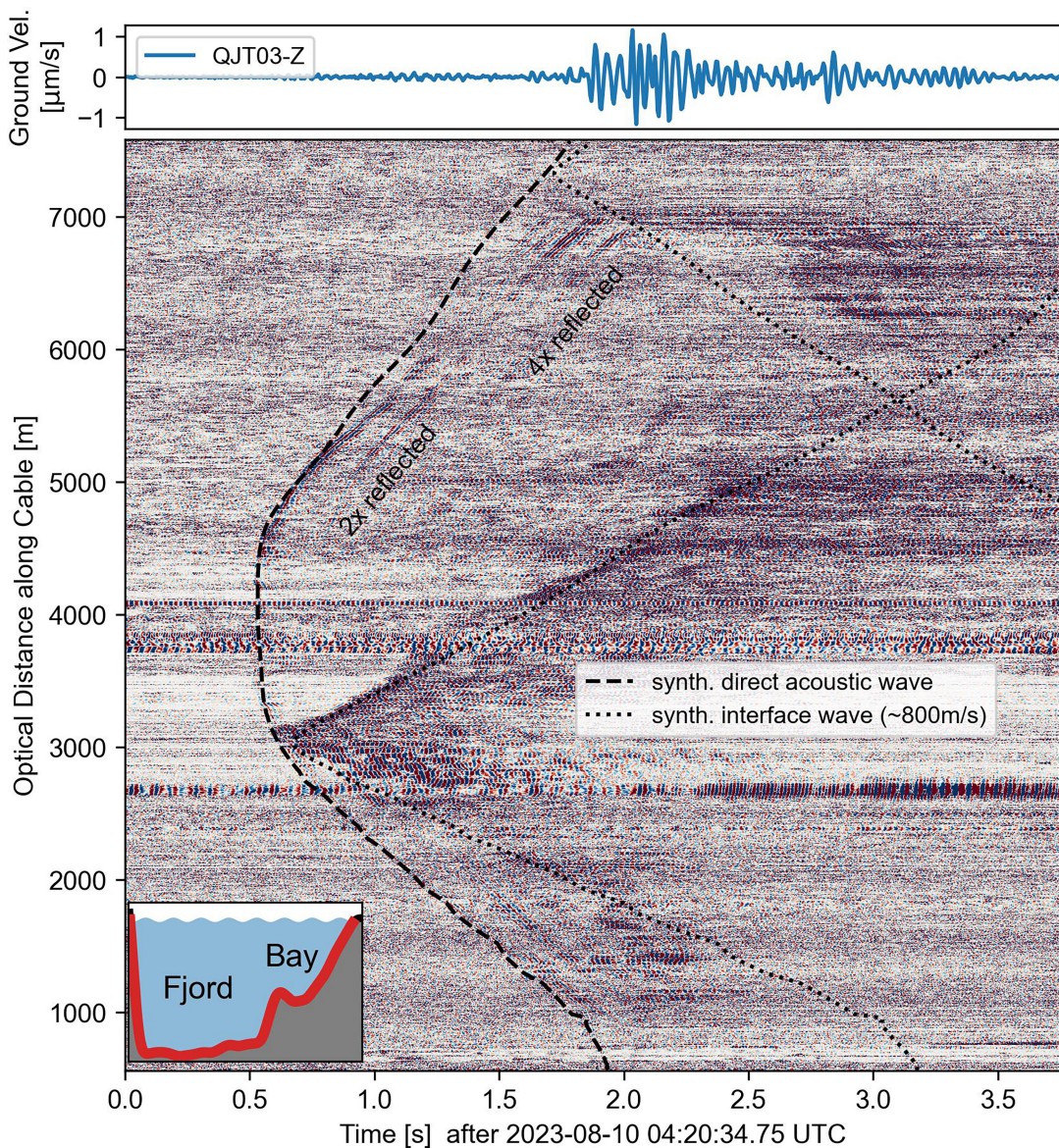

**Extended Data Fig. 3 | Iceberg fracturing recorded on the full cable length.** DAS strain record (normalized per channel) of an iceberg fracturing event south of the cable at 61.2993, −45.7772 (latitude, longitude). Dashed lines indicate simulated arrival times for acoustic waves and dotted lines for converted interface waves. Vertical seismogram of QJT03 at the top. This event was recorded when the entire cable was operational.

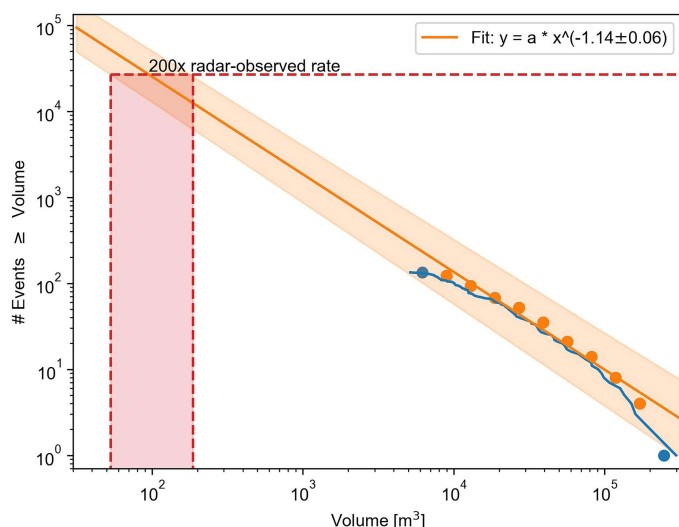

**Extended Data Fig. 4 | Radar-observed calving statistics and derived DAS calving-volume sensitivity.** Calving event volume-number statistics from TRI observations (blue line and all scatter dots) showing the number of events larger than a given volume on the horizontal axis. A 'roll-off' is evident below 10,000 m³ due to the limited resolution of the TRI. This means that the observed rate for events below 10,000 m³ is smaller than expected from self-similarity. For larger volumes, the calving event statistics follow a power-law as expected for a self-organized criticality (orange fit to observations with $1\sigma$ uncertainty as orange shading, only orange scatter dots taken into account). Above 100,000 m³, self-similarity of calving-event volumes breaks due to the finite size of the calving front and short observation time. With DAS, we observe 200x more calving events than with the TRI (upper red dashed horizontal line). By extrapolating the radar-derived power law (orange line) to event rates 200x larger than the TRI, we estimate the DAS detection threshold to about 50–200 m³ (red range projected on the horizontal axis indicating the expected volume within $1\sigma$ uncertainty).

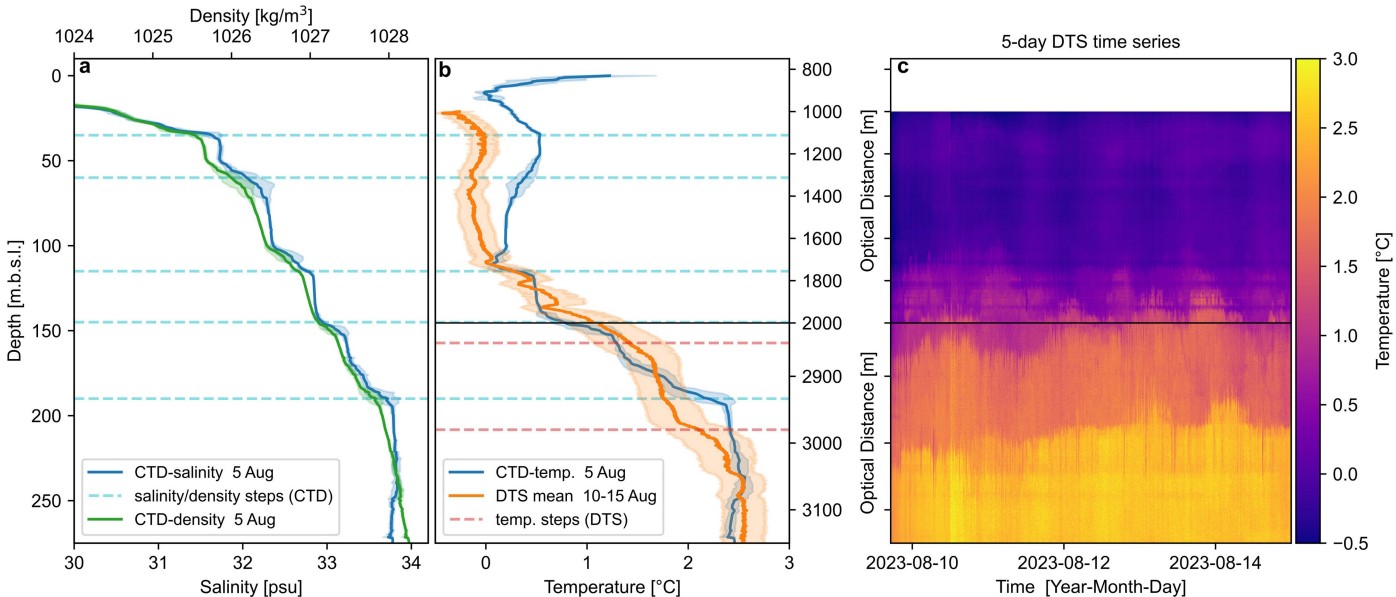

**Extended Data Fig. 5 | CTD profiles compared to DTS time series. a**, Salinity and density, averaged from 4 CTD profiles close to where the fibre-optic cable enters the deep fjord from 5 August. 1σ uncertainties are represented by shaded regions around the average. **b**, Temperature from the same CTD profiles and DTS mean between 10–15 August. Depths with large temperature gradients are shifted because of two reasons: The OD to depth relation is not linear, and the CTD and DTS measurements are 5–10 days apart from each other. Note that above 150 m depth the DTS measures in the shallow bay away from the calving front. **c**, 5-day DTS time series illustrating the variance in the location of the thermohaline staircase steps.

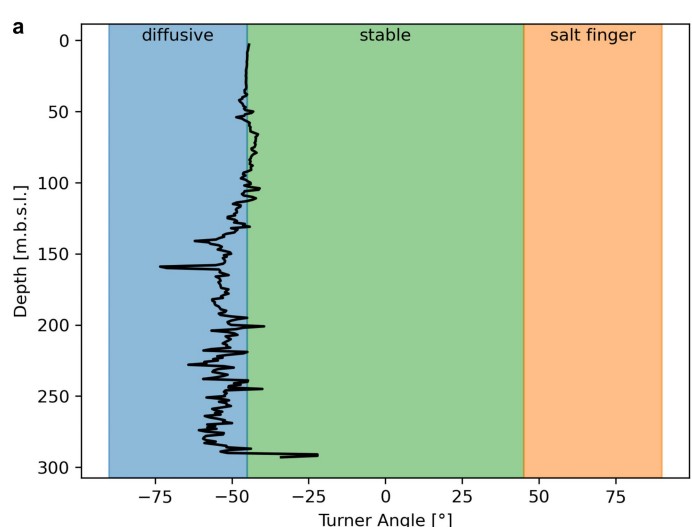

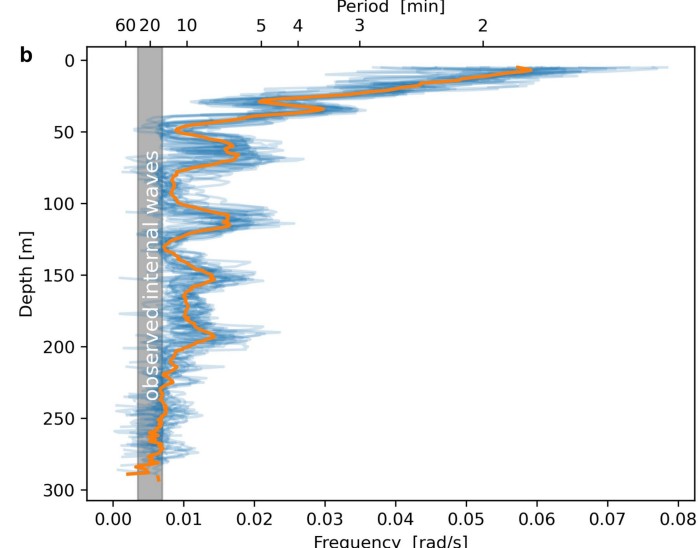

**Extended Data Fig. 6 | CTD-derived double diffusivity regime and buoyancy frequency. a**, Turner angle derived from 22 CTD profiles along the calving front. A Turner angle between −45° and +45° marks a stable stratification. For smaller Turner angles, diffusive convection will occur, whereas for larger Turner angles salt fingering will be present, both causing double diffusive convection. **b**, Brunt-Väisälä frequency (buoyancy frequency) derived from 22 CTD profiles along the calving front with 150 m spacing (blue). Orange line shows the mean of all blue lines. Gray shaded region indicates the band (15–30 min) in which we observe internal waves. Our observed frequencies are below the buoyancy frequency, which means that they can propagate and therefore are non-evanescent.

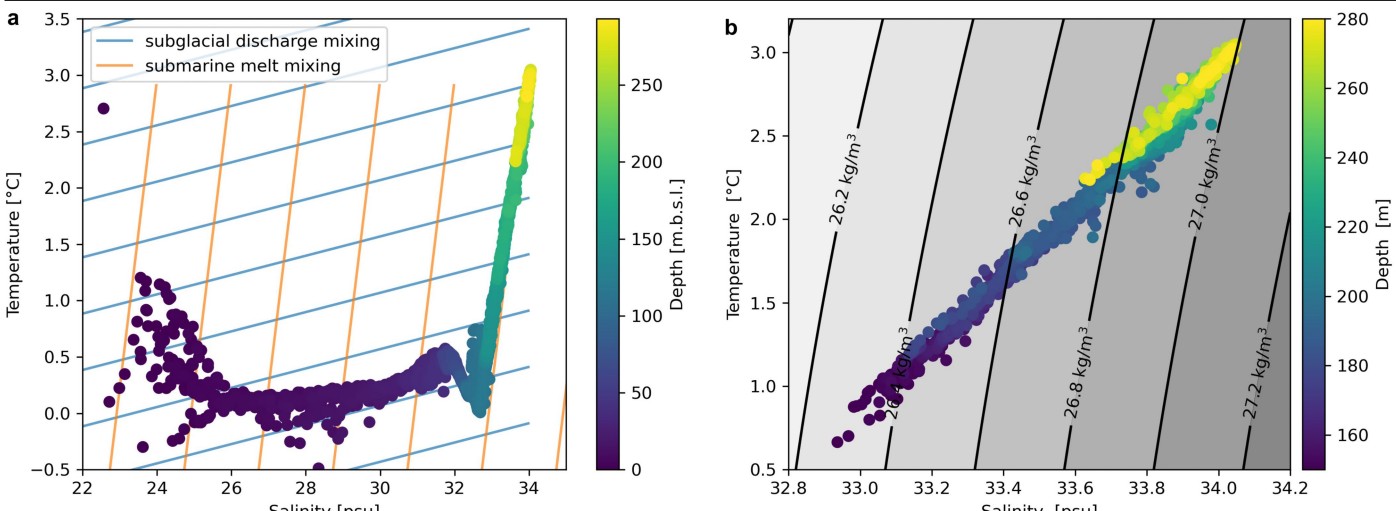

**Extended Data Fig. 7 | CTD-derived temperature-salinity dependence.**
**a**, Temperature-Salinity plot with depth colour-coded from CTD profiles along the calving front. Subglacial discharge modified waters are found above about 100 m depth. Subglacial discharge mixing lines have slopes of $[T_d - T_0]/[S_d - S_0]$ with subscripts $d$ and 0 denoting the deepest waters and surface waters respectively. Submarine melt mixing lines have slopes of $[T_d - (-90\,°C)]/[S_d - 0]$, with $T=-90\,°C$ the effective temperature of submarine melt that accounts for latent heat that is extracted from the ocean. **b**, Water density dependence on temperature and salinity between 150–280 m. Calculated density anomaly from 1,000 kg/m³ for 0 bar reference pressure. Density is only weakly dependent on temperature but strongly on salinity below 150 m depth, where we measure IGWs. This means that the water density is salinity-dominated in our case. Salinity scales linearly with temperature $S = 0.5\,psu/\,°C*T + 32.6\,psu$.

**a**

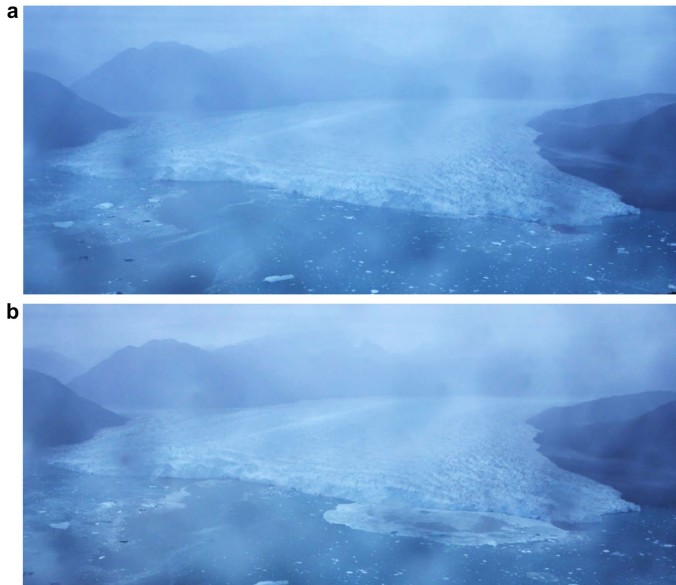

**b**

**Extended Data Fig. 8 | Timelapse imagery of calving event causing internal wave train.** Time lapse images from the camera south of the calving front. **a**, 2023-08-24 06:57 UTC, **b**, 2023-08-24 07:17 UTC This calving event caused the IGWs shown in Fig. 6a/b/c/d of the main text.

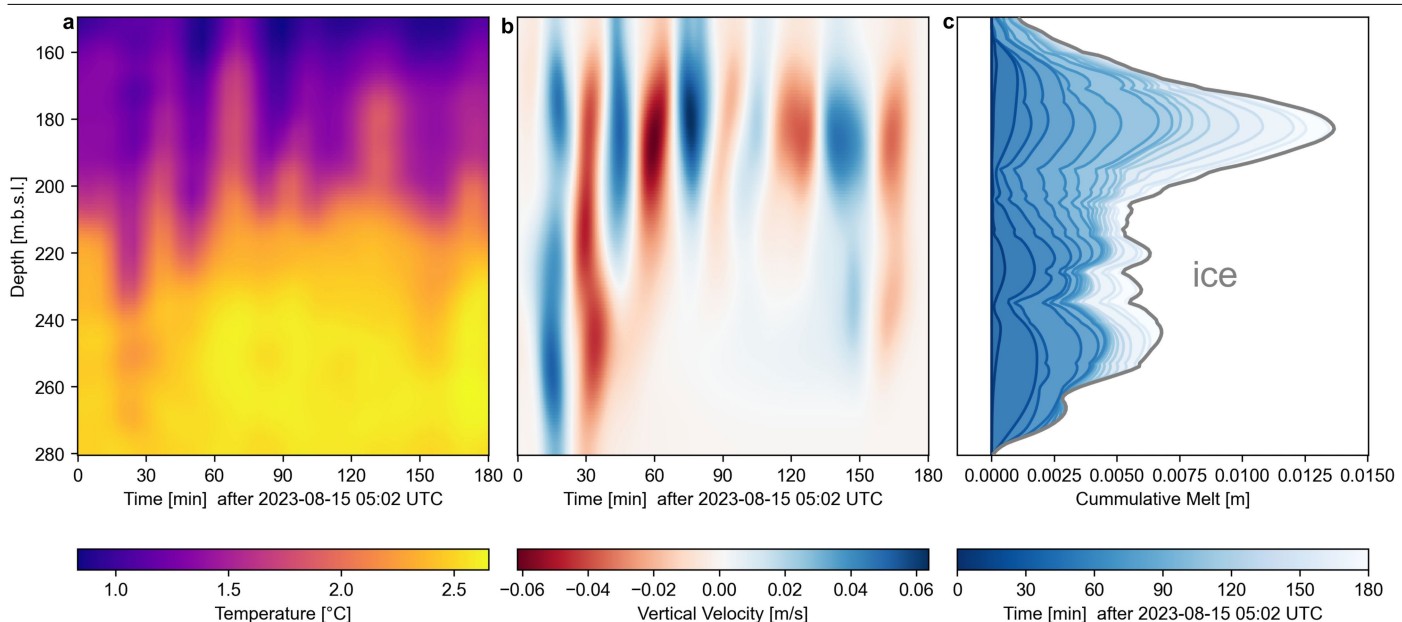

**Extended Data Fig. 9 | Submarine melt enhancement from internal waves.**
**a**, 3-h-long DTS temperature time series during the internal wave train passage of the largest observed calving event. **b**, Vertical velocity of water column during internal wave train passage. **c**, Cumulative submarine melt caused by turbulent heat transfer. Contour lines in 10 min intervals. Only the vertical velocity component is taken into account here.

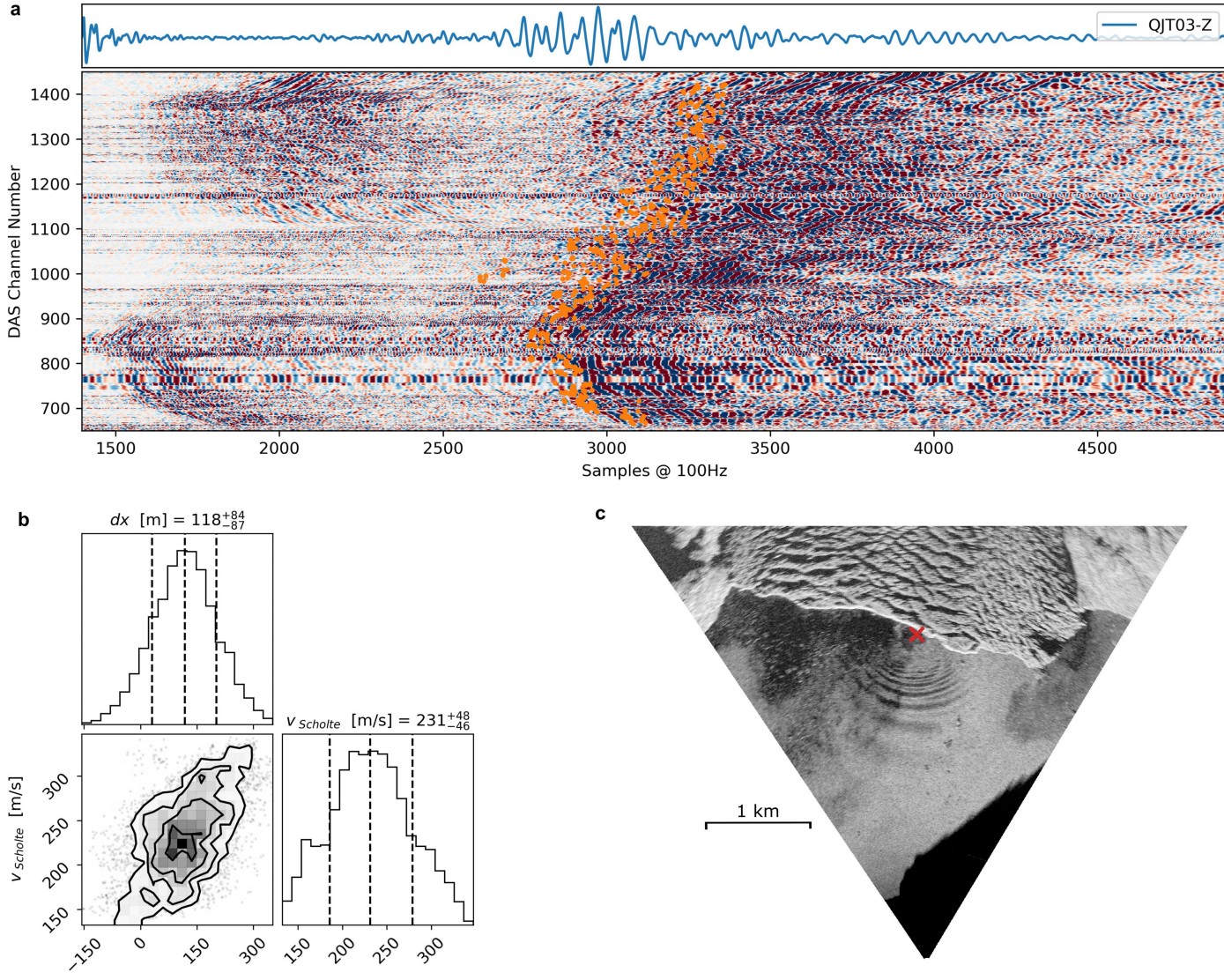

**Extended Data Fig. 10 | Cable layout inversion. a**, DAS strain data (normalized per channel) showing the Scholte wave excited by a calving event. Orange dots indicate arrival time picks from a 1 s /10 s STA-LTA trigger. The distribution of picks is treated as a probability distribution in the Bayesian inversion for the cable location. **b**, Distributions of walkers from the Monte Carlo Ensemble sampler for inversion parameters: dx: relative East-West calving event location, v: Scholte wave speed. **c**, Orthorectified TRI backscatter image of the ice front with calving-induced tsunami wave. This calving event was used to invert for the cable location in the deep fjord. Red cross marks the calving event location.

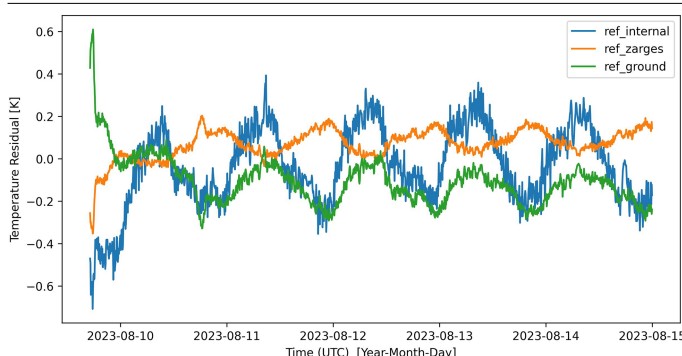

**Extended Data Fig. 11 | DTS calibration residuals.** Temperature from PT100 sensors subtracted from the calibrated DTS temperature at the corresponding reference. The internal (blue) reference section was not used for calibration. A 24 h oscillation is evident and results from slightly different temperatures at the PT100 sensors and along the fibre in the reference section. In the DTS data that we present, we correct for this artifact. We extract the oscillation in the bay cable section (1,000–1,800 m OD), where we do not expect strong (≳0.2 K) diurnal oscillations, and subtract them across all channels.