## [Peer Review File · Nature]

Calving-driven fjord dynamics resolved by seafloor fiber sensing

Corresponding Author: Dr Dominik Gräff

Version 0:

Reviewer comments:

Referee #1

(Remarks to the Author)

Review of Gräff et al. Manuscript

Gräff et al. used submarine fiber sensing in a remote environment near the tidewater glacier of Eqalorutsit Kangilliit Sermiat in Greenland. The high spatial resolution of this dataset allowed them to analyze several glacier calving processes that are otherwise hard to examine with land-based seismometers. As seismologists, we focus on the DAS observations in our review.

Major points:

The observations from DAS recordings are remarkable and very exciting. They allow for novel, detailed and fundamentally unique insights into glacial and sea interactions and are truly multi-disciplinary.

Overall, we believe the manuscript could be greatly improved through revision and additions to the figures and figure captions. Each figure has an enormous amount of data and detail, but they are all challenging to decipher as presented. Due to this and some lack of detail and consistency in the language within the text and the figure captions we find that the manuscript is difficult to follow at times. We provide feedback on each figure below in order to improve the presentation and the readers' understanding of the setting, data and results.

Additionally, we feel that the main text could have some additional quantitative elements to support the statements. For example, the total number of detected calving events over the time of the deployment as well as the ratio between submarine and subaerial.

The citations of the figures within the last ~25% of the text and annotations of particular signals within the DAS and DTS records would greatly aid the reader in understanding the observations and interpretations. Suggestions for these are given below.

Minor/moderate points:

L 103 - 104: Is this statement fully accurate? For example, <https://www.science.org/doi/full/10.1126/science.adp8094>, while not submarine fiber sensing, shows that DAS has been used for understanding ice stream dynamics. Consider softening the claim with 'to the best of our knowledge' or clarifying the specific gap.

L 107 - 110: The second part of sentence could be removed.

L 145: (upper panels)?

L 170 - 173: Slightly too long sentence here.

L 187 - 189: How many iceberg detachments were detected in total and how many of these were submarine and subaerial?

L 189: Not sure I follow what "therefore lacks these" refers to.

L 216: Quantify several.

L 217: Tide gauge can be included in Figure 1 (see comments for Figure 1 below)

L 223: Can you annotate this feature on Figure 4a

L 230: write out in full SGW

L 245: define CTD

L 263: 0:00 is that pm or am? Can you annotate with shading or an arrow in Figure 5a?

L 263-265: Does this need a reference?

L 265: Comma before where?

L 272: How were the vertical displacements measured?

L 279 - 285: This part I find hard to follow. Not sure what ~5 m/d refers to.

L 296: Saturate in amplitude? Measuring strain? How does this work?

L 306-307: Where is this observed? Can you cite the figure where this is shown?

L 311: Cite the figure where this is observed.

L 332-334: What is the delay time in these two observations? Can you calculate the velocities?

L 336: Cite the figure where this is observed.

L 343: Images of the sediment cover are not described or quantified. This may be over stating unless these can be cited and quantified.

L 354 - 357: Maybe remove "in"

L 478: I would suggest that some of these numbers communicated here to be included in the main text (e.g., 133 vs 10,000 calving events and the threshold of detection with DAS) as they highlight the high resolution of the DAS dataset.

Figures:

General: Perhaps you should also mention in the methods section how and why some DAS figures are plotted in strain-rate and others are in phase.

Figure 1.

What are the two types of white lines in (a). Could you please clarify this either in caption or a legend.

Why not make the map slightly bigger to show QJT02, which would also help orient the reader to the study region (similar to Figure S4 for example).

Maybe change the color scale for the OD as it is very similar to the satellite image of the water (and ice) in the fjord.

Clarify caption – the wording is difficult to follow. Remove "(pyGMT)" from caption and add a citation for it where appropriate (e.g., acknowledgments).

Figure 2.

Scales missing from both DAS and seismograms.

Were the more distant channels (> 5000 OD) not recording (aka broken as described in the methods)? If this is the case, then please add to the caption. Otherwise, why aren't you plotting the rest of the signals similar to in Figure 3?

The cyan dotted line is hard to see.

It would be great to add some of the OD information in the inset if possible, so the reader does not have to refer to figure 1.

Figure 3.

Scales missing from both DAS and seismograms.

What is the dispersive wave packet observed in the QJT03 seismogram in (b)? It is not observed in the DAS records, correct? Or am I misreading the alignment of the synthetic arrival in the DAS vs. the synthetic Scholte wave. Is this the conversion of Scholte waves to surface waves? In the text at the top of page 6, you state these are not observed in terrestrial seismometers – so I'm unsure what this signal is. Maybe it needs annotation or description if this is a different calving event signal or is the converted surface waves (which it may be ...)

Figure 4.

1/15 and 1/30 Hz are unusual representations of frequency. Why not use period? What is the corner frequency of this interrogator?

(a) What is the scale for the orange line representing the DAS 50m stack? Can you annotate the location of the channels stacked in the top waterfall plot? Can you annotate where the tide gauge is located or nearest in the waterfall plot?

(c) Why is the spectrogram from 30 m stacked DAS? Why not use 50m to be consistent with the stacked waveform in (a)?

Why didn't you determine the synthetic tsunami arrival based upon the bathymetry rather than a constant 300 m depth? Where and/when is the origin of tsunami wave? Could you annotate this somewhere clearly to help the reader?

Annotate the location of the subsea ridge that is described on page 6.

Figure 4 caption:

How many channels are stacked in the 50 m stacked line in (a)?

What is the 'deep water limit'? Can you define?

Not sure 'The phase offset between both originates from DAS approximately measuring strain-rate, proportional to the time derivative of water pressure at relevant frequencies.' is necessary in the caption. Perhaps this should be stated (and expanded upon) within the text.

What is the pining point island?

Revise last sentence 'Calving event location marked with a red cross at lat/lon as also shown in Figure 1b.'

Figure 5.

For (c) and (d) why did you plot the raw phase record? Why not strain-rate?

Annotate the impulsive wave packets identified in the text.

Annotate the IGW propagation and oscillations

Annotate the transient overturnings

Figure 5 caption.

Note the depth scale in (a) in the caption.

Needs descriptions of annotations of the primary observations (similar to the grey panel)

Figure 6.

(b) Annotate the IGW waves in the DTS.

(e) scale is flipped in comparison to figures a-d. Is this on purpose? It makes it difficult to read and compare to the other subfigures.

Figure 6 caption:

Note that the locations of the fiber (OD) are shown schematically in the insets. The insets are not mentioned in the caption.

(c) DTS deep seafloor temperature anomaly – explain this differently and note the OD position of each.

Referee #2

(Remarks to the Author)

I co-reviewed this manuscript with one of the reviewers who provided the listed reports.

Referee #3

(Remarks to the Author)

Processes at the glacier/ocean interface are poorly understood and largely under-observed, with implications regarding our ability to understand any interaction of calving, submarine melting and the fjord circulation. This missing physics means that these processes are not adequately represented in models and a contributor to uncertainty in projections of ice loss and the associated sea level rise or freshening of the oceans.

This study presents measurements collected using novel systems including a DTS (distributed temperature sensing) and DAS (distributed acoustic sensing) at the edge of a medium-size marine terminating glacier in southwest Greenland. The power and novelty of these systems is the ability to record variability on a wide range of scales (from milli-seconds to days) generated throughout the entire cycle of icebergs formation and drift: initial fracturing, detachment and subsequent drift away from the calving front. The authors convincingly describe the various types of waves (or wave-like processes) associated with these events including seismic waves from the fracturing, waves excited at the water/sediment interface (Scholte waves), tsunamis (transient surface gravity waves) excited by iceberg calving, and internal gravity waves in the fjord – including those present in the wake of the iceberg. Through several key examples for each type of waves, they show that the combined DTS and DAS systems can successfully record these waves and provide information on their propagation, amplitude and origin location.

Given the ability of these systems to record both the calving and the waves generated by calving events – the authors then speculate that these systems could be used to track calving events, separate aerial and submarine calving events, quantify calving volumes, advance detection of tsunami waves and quantify the impact of internal gravity waves on mixing and submarine melting. Furthermore, they conclude that this study provides evidence of a potential feedback where calving excites waves, which increase melting at the ice/ocean boundary, which then impacts calving.

The novelty of the paper is that it demonstrates that DTS and DAS sensors can be successfully deployed in front of calving glaciers and that, when deployed, they record a multitude of transient wave-systems generated throughout the calving cycle. Some of these processes had already been recorded using different observing techniques (eg surface gravity waves and location, Minowa et al. 2018, or internal gravity waves, Cusack et al. 2023) – the novelty here is in demonstrating that they can all be recorded by this system, including over larger distances and at much higher resolution/information. From a science result perspective, showing that many of these processes are occurring (and hence interacting) at the same time is novel. Some of the sediment/ocean waves and the impact of waves on temperatures at the seafloor are also novel. The observations and analysis techniques described here have the potential to revolutionize the study of the glacier/ocean interface.

What this study does not do is present new findings that use the observations made to quantitatively assess the impact of e.g. one particular wave-type on a glacial process. For example, the authors use vertical velocity data, together with other fjord collected observations, to show how submarine melting could be estimated but admit that more data and more analysis would be needed for a more realistic estimate. Similarly, they argue that these systems may be useful in early detection of tsunamis, generated by calving, which may pose a threat to people living in the vicinity of the glacier. But much more work and planning would be required to determine the feasibility of such a system.

The paper is well written, the analysis presented is sufficiently detailed and explained.

In summary, the strength of this study is in the novelty of the methodology and a demonstration that it can record a wide range of wave-types associated with calving processes. This opens the door to the deployment of this system in different glacial environments to quantify and describe these different processes in greater detail. As such it is an extremely valuable step forward in our understanding of how to observe these challenging systems.

Referee #4

(Remarks to the Author)

Dear Nature Editors and Manuscript Authors,

In this study, Gräff et al describe observations conducted near a major south Greenland glacier (Eqalorutsit Kagilliit Sermiat) using a fiber optic cable (and other instruments). Their measurements detect many of the processes known to cause ice loss from tidewater glaciers (e.g. “subaerial” calving, internal waves) as well as some processes that have been largely unobserved (e.g. submarine calving, iceberg currents). The observations reveal a “chain” of processes that contribute to ice loss from the glacier. This study is timely and of scientific interest because frontal ablation-induced ice loss from Greenland’s glaciers leads to a substantial portion of mass loss from the ice sheet and global sea level rise.

The use of the fiber optic cable for this application is extremely novel and the integrated observations from this field campaign are likely to be of widespread interest to the glaciological and oceanographic communities and beyond. In reading this manuscript, I had two major comments described here as well as a number of line and figure suggestions, appended below.

First, given that the fiber optic cable is the novel new observational instrument in this study, there is a lot of missing information that the reader needs to interpret how this instrument works and how the authors have made their conclusions. As of now, there is a major leap in logic that is hard to follow. One main update that would help readers is a paragraph circa line 112 that describes how the fiber optical cable is sensing changes in the water (e.g. is it expanding and contracting? something else? what is a channel? why is phase important? etc) and how these signals are sensed, parsed, and eventually interpreted after collection. I don’t think the paragraph should be highly technical (although perhaps some more technical details summarizing the measurements should be placed in the methods) but some sort of description is necessary. This would also really help the reader in interpreting many of the main figures (e.g. 2 and 3) which are currently lacking in units and even colorbars, obscuring a clear understanding. I get the sense that the authors may have looked at these types of figures long enough that they are easily interpretable to them but that is not the case for the standard interested scientist.

Second, one of the main claims and selling points of the paper is that the observations from the fiber optical cable help “close the ice front ablation budget” (Line 72) and indeed a lot of novel processes are identified. However, the ice front ablation budget is never explicitly calculated. This calculation would significantly improve the impact of the manuscript and would help make the case that this type of instrumentation is a breakthrough in assessing ice sheet mass balance. Thus, I think it is necessary to attempt a budget calculation, even if the uncertainties are large, so that readers can understand what processes are “important” and where focus should be placed for future studies. This calculation would also make this paper a blueprint for future studies with fiber optic cables near glaciers. In addition, it would also help support the authors expectation that this will be applied in diverse studies at other fjord systems (line 353).

Overall, I recommend this paper for publication in Nature after addressing the major comments above. Please don’t hesitate to reach out to me if you have any questions about this review.

Sincerely,

Mike Wood

Line comments:

2: I might suggest “revealed” in place of “illuminated” since the cable is not an active (vs passive) sensor (...right?)

82: Should probably define GrIS here.

94: strongly affect calving activity. > strongly inhibits calving in winter in systems with especially thick mélange.

97: re: thermal structure: Davison et al 2020 show the effect of icebergs on thermal structure so this is not totally “hidden”

112: I might also include being hidden at depth away from icebergs as a benefit

117: 0.3% of the GrIS area

145-146: This is a little confusing – are there not “wiggles” in the plots above Figure 2 at the same time as the wave arrival? Units and a description of these subplots in Figure 2 would be helpful (see comments on figures below).

150: Why would water-sediment waves influence ice front stability? This seems speculative without an explanation.

160: moment -> timing

168: remove “into”

172-173: If submarine calving events are not attached to the sea floor or fjord walls (and therefore do not provide buttressing) then do they really matter in models for long term evolution of glaciers and ice sheets? If so, this needs an explanation.

180-181: probably need a citation here

188-189: former -> latter? Submarine detachments would cause a cavity, correct?

227: vertices? (epicenters is plural)

239: What is the depth of the internal waves/thermocline? This will dictate the extent to which they impact melt. (e.g. are the IGW-induced temperature variations and water velocities mostly near the surface? Or at depth where the glacier is more vulnerable?)

242: vertical water velocities?

249-251: There is a lot of jargon in these lines for the general reader (Kelvin-Helmholtz instabilities, Turner angles). Could this be said a different way here with some of the details put into Methods for the interested reader? I see Turner angles are in the Methods already.

264: breaking

264-265: This seems a little speculative without additional information

271: to calving events -> as originating from calving events

297: what is a “hyperbolic move out”?

301: illuminate -> reveal (same comment above)

301-304: Do I understand this section right that an internal wave travels back toward the glacier from the calved iceberg that is floating away? If so, it would be good to explicitly mention this.

337-338: I think this needs to be qualified/nuanced a little. Ice mélange is shown to contribute to buttressing during winter at very large glaciers with thick, jam-packed icebergs in a fjord. I doubt this process is at play at EKS.

341: remove bespoke

352: what is a wide spatial aperture and why is it important (not mentioned until here in the conclusions)

373: the IGWs are induced by calving but calving is most prominent in the melt season – why would it be most important outside the melt season?

375: Were there observed major calving events in the Sutherland et al study between the front scans that led to this 2 orders of magnitude conclusion?

Figure Comments:

Figure 1: Red rectangle -> Green rectangle? Also, I think “orthophoto” should be “imagery”.

Figure 2: This figure is unclear because it is missing a lot of information. Some questions: What are the units on the top panels? What are the colors in the plots (blues and reds)? What are the units? What are the horizontal variations around 2000 and 2600 m? If QJT01 is closer to the Bay, then why does it receive a signal before QJT02, which is closer to the calving event? Also, it's difficult to see red and blue dots on a red and blue figure.

Figure 3: Suggest labeling calving events explicitly with annotations and text

Figure 4: This figure is also unclear. What is a “DAS 50m stack”? What do the colors indicate in panels b and c? What are the units? Are they the same (if not, perhaps a different colormap for each would be a good idea)?

Figure 5: Some selectively chosen isotherms on panel a would help the reader identify which features are important in this figure. Also, what is happening in the transition just after 2023/8/27 21:00?

Figure 6: What is the temperature anomaly in reference to? What is the difference between b and d (just the scale)? What is a DAS Channel (not described in text, see first major comment).

Figure 7: This is a nice schematic and I feel like it should come earlier in the manuscript. I realize all the text builds up to this

figure but since it encapsulates the “story” perhaps it could be moved sooner? Also, since this is a “chain” of events, it might make better sense for these to be labeled with numbers (Step 1, Step 2, ...) rather than letters (which are used for subplots in the previous images).

Version 1:

Reviewer comments:

Referee #1

(Remarks to the Author)

The authors have addressed most of our and others' first review round of comments. The revised manuscript is slightly more concise, and the changes to both the figures and the text have improved the readability of the article up to a point. The authors have made efforts to quantify more of their observations (example L295-298), which is helpful. However, the track changes document mostly highlights the minor changes due to citation reordering and figure reordering. There are only a few substantial modifications to the text that in our opinion reflect how the constructive comments and suggestions given by the 4 reviewers were addressed.

Some positive feedback is that we find moving the illustration that describes the processes as Figure 2 helps. However, the text frequently refers to figures out of sequential order, which makes it hard for the reader to go back and forth. We are afraid we do not have any new useful comment on this, but a more logical and consistent way to align text and figure order would further enhance the clarity.

Minor/moderate points:

L 125: Consistent use of fiber and fibre?

L 156: Would 'travel times' be a better fit rather than 'time of flight'?

L 190: There is no blue curve on Figure 3b – perhaps the authors mean 3d.

L 279: Period missing.

L 391: “Submarine melt” appears twice in the same sentence. Maybe replacing “submarine melt models” with glacier models would be better phrasing.

Figure 4 caption (previous reviews):

The authors misunderstood 'define'. Please describe/note/quantify the number that represents the “deep water limit” in the caption. We did not request the equation, just a number to help the reader understand the figure.

Regarding the pinning point island comment – that is why we asked. It is not in the caption, yet it is referred to in the text when describing Figure 4. Please add or modify to clarify.

Figure 6(e). This doesn't seem important enough to be shown in the main text. Perhaps consider moving it to the Extended Data.

Referee #2

(Remarks to the Author)

I co-reviewed this manuscript with one of the reviewers who provided the listed reports.

Referee #4

(Remarks to the Author)

Dear Nature Editors and Manuscript Authors,

This is my second review of Gräff et al's manuscript describing observations conducted near a major south Greenland glacier (Eqalorutsit Kagillit Sermiat) using a fiber optic cable and other instruments. In response to comments from me and the other reviewers, this manuscript has undergone significant revisions that improve the overall presentation fo study.

In my previous review, I had two major comments, which I outline here along with my summary and impressions the authors' response.

My first comment pertained to the description of the DAS/DTS systems and a lack of introductory information on how the system works. The authors have now provided 4 sentences in their introduction in the manuscript about this system which greatly improves the reader understanding of what the system is measuring and what some of the jargon means. This is sufficient to address the point I brought up (and I certainly didn't intend for a full description of the physics of the measurement system at this point in the manuscript). I also noted in this comment that many of the plots were missing

colorbars. Some colorbars were added to a few plots and the authors have elected to describe the blue/red compressional/extensional strain in the caption of figures without colorbars.

My second comment pertained to closing the ablation budget, which was not quantified yet was indicated as an important outcome from the measurements presented in this study. The authors have now updated the manuscript to provide an estimate of terms that are possible to robustly quantify with their measurements. In addition, they have convincingly described some of the reasons why a full closure of the budget is not possible with their sensors in the response to reviewers.

In addition to the major comments above, I also provided several line and figure comments and suggestions. The authors have gone through each of the suggestions and provided, where possible, appropriate updates to the manuscript and/or responses to each of these points.

Overall, my comments have been addressed and I recommend this manuscript for publication in Nature.

Sincerely,

Mike Wood

Comments of Referee #1 and Response:

Gräff et al. used submarine fiber sensing in a remote environment near the tidewater glacier of Eqaqorsut Kangilliit Sermiat in Greenland. The high spatial resolution of this dataset allowed them to analyze several glacier calving processes that are otherwise hard to examine with land-based seismometers. As seismologists, we focus on the DAS observations in our review.

Major points:

The observations from DAS recordings are remarkable and very exciting. They allow for novel, detailed and fundamentally unique insights into glacial and sea interactions and are truly multi-disciplinary.

Overall, we believe the manuscript could be greatly improved through revision and additions to the figures and figure captions. Each figure has an enormous amount of data and detail, but they are all challenging to decipher as presented. Due to this and some lack of detail and consistency in the language within the text and the figure captions we find that the manuscript is difficult to follow at times. We provide feedback on each figure below in order to improve the presentation and the readers' understanding of the setting, data and results.

We thank referee #1 for the positive feedback. We agree on the points made about the complexity of the figures and discuss our adjustments below, in-line with the detailed comments on the figures.

Additionally, we feel that the main text could have some additional quantitative elements to support the statements. For example, the total number of detected calving events over the time of the deployment as well as the ratio between submarine and subaerial.

Done. We added: *"In total we detected ~56,000 iceberg detachments in the DAS data, of which ~35,000 had higher-order Scholte wave signals above the noise floor in the bay, and therefore setting a lower limit on the submarine-to-subaerial calving event ratio of 1/3."*

The citations of the figures within the last ~25% of the text and annotations of particular signals within the DAS and DTS records would greatly aid the reader in understanding the observations and interpretations. Suggestions for these are given below.

We clarified with more references in the text and annotations in the figures.

Minor/moderate points:

- L 103 - 104: Is this statement fully accurate? For example, <https://www.science.org/doi/full/10.1126/science.adp8094>, while not submarine fiber sensing, shows that DAS has been used for understanding ice stream dynamics. Consider softening the claim with 'to the best of our knowledge' or clarifying the specific gap.
We believe that this statement is correct. Yes, DAS has been used on glaciers and ice-streams before¹⁻⁶, but not in the context of ice-ocean interactions. Unfortunately, due to the limit on the number of references, we cannot include these studies.
- L 107 - 110: The second part of sentence could be removed.
Done.
- L 145: (upper panels)?
Corrected.
- L 170 - 173: Slightly too long sentence here.
Split into two sentences.

- L 187 - 189: How many iceberg detachments were detected in total and how many of these were submarine and subaerial?
Good point. We detect 56000 iceberg detachments from Scholte waves in the fjord. 35,000 of these were also detected by their higher order Scholte waves in the bay that are possibly scattered from acoustic emissions. However, that number depends on the association of triggers in the deep fjord and in the bay, which is challenging for iceberg detachments that are only a few seconds apart from each other. We added: *"In total we detected ~56,000 iceberg detachments in the DAS data, of which ~35,000 had higher-order Scholte wave signals above the noise floor in the bay, and therefore setting a lower limit on the submarine-to-subaerial calving event ratio."*
- L 189: Not sure I follow what "therefore lacks these" refers to.
We clarified by: *"...and therefore lacks these acoustic emissions and the cascading higher-order Scholte waves that are excited by them."*
- L 216: Quantify several.
Good point, which is not so easy to answer, because the tsunami signals often overlap. This is due to multiple large calving events exciting tsunamis within a few minutes and due to tsunami reflections in the fjord and the bay. The other problem here is a simple definition problem: What characterizes a tsunami? This becomes relevant for smaller calving events producing smaller tsunamis close to the background wave height from excitation by wind gusts. This makes it hard to quantify the number of tsunamis during our deployment. As an example, we show the tide gauge data combined with DAS data in Fig. R4 of this document. To quantify the number of tsunamis during our deployment we trigger on the tide gauge data, because this works more robust than the triggering on the DAS data, resulting in 600 tsunamis during our deployment time. The reason that the triggering works more robust on the tide gauge than on the DAS data is that drifting icebergs cause huge signals in the DAS record (even in the bay), limiting the sensitivity of the trigger. We included in the manuscript: *"We observed several hundred tsunamis during our study period ..."*
- L 217: Tide gauge can be included in Figure 1 (see comments for Figure 1 below)
Done.
- L 223: Can you annotate this feature on Figure 4a
Done.
- L 230: write out in full SGW
The abbreviation for surface gravity waves (SGW) is already introduced previously in the first sentence of the subsection. Therefore, we would like to use the abbreviation here and in the following.
- L 245: define CTD
Done
- L 263: 0:00 is that pm or am? Can you annotate with shading or an arrow in Figure 5a?
We use ISO 8601 for time formatting. Therefore, we would like to avoid using the American time formatting with 'am' or 'pm' to stay consistent with previous dates and times stated in the manuscript. We annotated the overturning.
- L 263-265: Does this need a reference?
Yes! We added: Ruddick, B. A practical indicator of the stability of the water column to double-diffusive activity. *Deep Sea Res. Part Oceanogr. Res. Pap.* **30**, 1105–1107 (1983).
- L 265: Comma before where?
Done.
- L 272: How were the vertical displacements measured?
Clarified by: *"The measured vertical isothermal displacements of the internal waves ..."*
- L 279 - 285: This part I find hard to follow. Not sure what ~5 m/d refers to.
We added *"This is comparable to the ice flow speed at the calving front during our observations (~4.5 m/d), ..."*
- L 296: Saturate in amplitude? Measuring strain? How does this work?
We assess the details to be too technical for the main text. The explanation was in the Supplementary Information and is now moved more prominently to the Methods section. The

reason is, that the phase unwrapping cannot be done accurately anymore, which results in a wrong phase measurement. We clarified by: *"DAS phase records typically saturate for these cooling events (see Methods) ..."*. We also now introduce the concept of DAS in third paragraph of the main text.

- L 306-307: Where is this observed? Can you cite the figure where this is shown?
Done. (Fig. 6f)
- L 311: Cite the figure where this is observed.
Done. (Fig. 6e)
- L 332-334: What is the delay time in these two observations? Can you calculate the velocities?
The delay time is ~ 1.2 s. With a straight-line source-receiver distance of 5,500 m, this results in a wave speed of 4,500 m/s. We added this to the manuscript: *"... traveled through the faster ice or bedrock with $\sim 4,500$ m/s."*
- L 336: Cite the figure where this is observed.
Done. (Fig. 3b along the dotted black line)
- L 343: Images of the sediment cover are not described or quantified. This may be overstating unless these can be cited and quantified.
We reformulated: *"we were able to characterize sediment cover and locate glacier pinning points on the seafloor landscape ..."*
- L 354 - 357: Maybe remove "in"
Done.
- L 478: I would suggest that some of these numbers communicated here to be included in the main text (e.g., 133 vs 10,000 calving events and the threshold of detection with DAS) as they highlight the high resolution of the DAS dataset.
Done. We include: *"... our seafloor DAS resolves and locates ten-thousands of iceberg detachments ... More specifically, our DAS-based calving catalog comprises 200x more detections and a 40% larger cumulative detachment volume than the TRI detected (see Methods, Extended Data Fig. 4)."*

Figures:

- General: Perhaps you should also mention in the methods section how and why some DAS figures are plotted in strain-rate and others are in phase.
We included in the 'DAS phase to strain conversion' section in 'Methods': *"For that reason, we show the DAS data in units of strain, raw phase record, or normalized per channel depending on the frequency of interest."*
- Figure 1.
 - What are the two types of white lines in (a). Could you please clarify this either in caption or a legend.
The white dashed line is explained in the first sentence. The white dotted line is labeled directly in the plot.
 - Why not make the map slightly bigger to show QJT02, which would also help orient the reader to the study region (similar to Figure S4 for example).
Extending the map to fit QJT02 on it produces lots of empty space in both N-S and E-W direction and would give less space for the profile. We think that the focus of this figure should be the cable layout relative to the calving front. We included an arrow that points in the direction of QJT02.
 - Maybe change the color scale for the OD as it is very similar to the satellite image of the water (and ice) in the fjord.
Done.
 - Clarify caption – the wording is difficult to follow. Remove "(pyGMT)" from caption and add a citation for it where appropriate (e.g., acknowledgments).

Done.

- Figure 2.

- Scales missing from both DAS and seismograms.

The seismograms and the DAS data are both normalized. We consulted the editor on the requirement to add colorbars to plots showing DAS data that has been normalized by channel. In his consent, we would like to refrain from using colorbars for plots showing normalized DAS data, as they would only show numbers between -1 and 1 but block figure space. However, we now labeled the seismometer axis (see also below for former Fig. 3). We also emphasized the normalization in the caption.

- Were the more distant channels (> 5,000 OD) not recording (aka broken as described in the methods)? If this is the case, then please add to the caption. Otherwise, why aren't you plotting the rest of the signals similar to in Figure 3?

Done. We added: *"We only plot ODs up to 5,000 m here, as this was the operational cable length at the time of the shown events."*

- The cyan dotted line is hard to see.

Replaced by black dotted and dashed lines.

- It would be great to add some of the OD information in the inset if possible, so the reader does not have to refer to figure 1.

We understand the point referee #1 makes here. However, given their critics that the figures are already loaded with information, we would like to avoid adding numbers of optical distance into the inset plots. We are now emphasizing that the cable section shown in each plot is colored red in the inset: *"The inset in the lower left corner shows an East-West cross section of the fjord, resembling Fig. 2c, with cable sections used for this plot marked in red."*

- Figure 3.

- Scales missing from both DAS and seismograms.

See above comment on former Fig. 2. We added scales for the seismograms, but refrain from using scales for the DAS data, as it is normalized for each channel.

- What is the dispersive wave packet observed in the QJT03 seismogram in (b)? It is not observed in the DAS records, correct? Or am I misreading the alignment of the synthetic arrival in the DAS vs. the synthetic Scholte wave. Is this the conversion of Scholte waves to surface waves? In the text at the top of page 6, you state these are not observed in terrestrial seismometers – so I'm unsure what this signal is. Maybe it needs annotation or description if this is a different calving event signal or is the converted surface waves (which it may be ...)

Correct, the dispersive wave packet observed at QJT03 is not visible (or at least not clearly) in the DAS record. It is the Rayleigh wave that travels from the calving event to the seismometer. It arrives there approximately simultaneously with our modeled synthetic higher order Scholte wave traveling at ~800 m/s, but that may be a coincidence and due to the limited plot resolution. In principle, it could also be an acoustic-to-surface conversion for a wave traveling first through the water and then coupling into the land. However, we have no evidence for that.

As pointed out, the conversion of Scholte waves to Rayleigh waves is visible at about 9s in Fig. 3b. To make it clearer we inserted *"(Fig. 3d orange line and simultaneous arrival at seismometer at ~9s)"* earlier. We also added in the figure caption: *"At ~9 s, the Scholte-to-surface converted wave arrives at QJT03."*

- Figure 4.

- 1/15 and 1/30 Hz are unusual representations of frequency. Why not use period? What is the corner frequency of this interrogator?

We would like to use the frequency representation for consistency but changed to 30-60 mHz to avoid having fractions.

This interrogator (Sintela ONYX v1 uses a leaky filter at 0.1 Hz that may be interpreted as a corner frequency. The effect of this filter on our signals was described in the Supplementary Information as we assess it to be very technical and will be moved to into the Methods section: "...The interrogator's internal data flow applies a leaky frequency filter at 0.1 Hz. For frequencies $\lesssim 10$ mHz, this filter acts as a differentiator, which results in phase measurements proportional to strain-rate, instead of strain."

- (a) What is the scale for the orange line representing the DAS 50m stack?
 - Good question. It is complicated. As the frequency content of the tsunami is around 0.1 Hz and therefore overlaps with the leaky filter, the recording is a mixture of strain and strain-rate. This is also true for the internal waves in Fig. 5c/d and Fig. 6b/d and is the reason why we report raw phase recording units instead of strain or strain rate.
 - Can you annotate the location of the channels stacked in the top waterfall plot?
Done. We added a color coding to the right of the plot a).
 - Can you annotate where the tide gauge is located or nearest in the waterfall plot?
The tide gauge is directly where the cable enters the water. We added to the caption: "The tide gauge is co-located with the lowest DAS channels in a." We also added markers to the right of Fig. 4a for the tide gauge and cable sections used in Fig. 4b/c.
- (c) Why is the spectrogram from 30 m stacked DAS? Why not use 50m to be consistent with the stacked waveform in (a)?
 - It is from a different cable section. The reason is, that the frequency content of the tsunami is strongly affected by reflections at the shore and therefore the wave dispersion is not so clear in the cable sections directly at the tide gauge. Another reason is that the cable direction close to the shore is oblique to the tsunami propagation (add modelled travel path), and therefore the DAS response is reduced. The 30m section at 1,000m OD has the clearest signal, as its ~500m away from the shore and the cable direction is parallel to the tsunami propagation. See Fig. R1 attached to this document.
 - Why didn't you determine the synthetic tsunami arrival based upon the bathymetry rather than a constant 300 m depth?
We wanted to keep it simple, but we agree with referee #1 to include it. We now simulated the tsunami arrival based on the bathymetry. As expected, it fits the data much better at shallow water depths.
 - Where and/when is the origin of tsunami wave? Could you annotate this somewhere clearly to help the reader?
We think we cannot make it much clearer than in Fig. 4d/e. Therefore, we added "... record of a calving-induced tsunami (location in d/e) filtered between ..."
 - Annotate the location of the subsea ridge that is described on page 6.
Done.
- Figure 4 caption:
 - How many channels are stacked in the 50 m stacked line in (a)?
10 channels. We added in parenthesis: "... alongside the 50 m (10 channels) stacked ..."
 - What is the 'deep water limit'? Can you define?
It is defined in the text. We would like to refrain defining it here, because it would require also defining variable like h: water depth, λ : wave length, g: acceleration due to gravity. We think that this is too much for a figure legend that is anyway already very long.
 - Not sure 'The phase offset between both originates from DAS approximately measuring strain-rate, proportional to the time derivative of water pressure at relevant frequencies.' is necessary in the caption. Perhaps this should be stated (and expanded upon) within the text.
Removed.
 - What is the pinning point island?

- The term “pinning point island” is not used in the caption.
 - Revise last sentence ‘Calving event location marked with a red cross at lat/lon as also shown in Figure 1b.’
Done.
- Figure 5.
 - For (c) and (d) why did you plot the raw phase record? Why not strain-rate?
See above comment. We are plotting a frequency range where the interrogator phase recording is a non-linear and unknown function of strain and its time derivative. Therefore, we cannot convert to strain.
 - Annotate the impulsive wave packets identified in the text.
Done.
 - Annotate the IGW propagation and oscillations
Done.
 - Annotate the transient overturnings
Done
- Figure 5 caption.
 - Note the depth scale in (a) in the caption.
Done.
 - Needs descriptions of annotations of the primary observations (similar to the grey panel)
Done.
- Figure 6.
 - (b) Annotate the IGW waves in the DTS.
This is very difficult to annotate as everything that is plotted are signals of IGW. We therefore try to clarify in the caption: “*Corresponding Distributed Acoustic Sensing (DAS) record, which saturates for the strongest IGW signals.*”
 - (e) scale is flipped in comparison to figures a-d. Is this on purpose? It makes it difficult to read and compare to the other subfigures.
Oh, thanks for that catch. We would like to use the convention from e), as it is also used in the fracturing and calving figures. We therefore swapped c) and d) and realized that c) was flipped with respect to d). Now the convention is: optical distances increase upwards, besides when a temperature profile is plotted, e.g. a) and b).
 - Figure 6 caption:
 - Note that the locations of the fiber (OD) are shown schematically in the insets. The insets are not mentioned in the caption.
Yes, but the inset is placed in every figure of the paper. Referring to the inset would cause a 4-fold repetition. We leave this decision to the typesetters.
 - (c) DTS deep seafloor temperature anomaly – explain this differently and note the OD position of each.
As the OD are clearer visible in (d) and to avoid cluttering, we simply add: “... (compare to (d))”

Combined Comments of Referee #2 & #3 and Response:

Processes at the glacier/ocean interface are poorly understood and largely under-observed, with implications regarding our ability to understand any interaction of calving, submarine melting and the fjord circulation. This missing physics means that these processes are not adequately represented in models and a contributor to uncertainty in projections of ice loss and the associated sea level rise or freshening of the oceans.

This study presents measurements collected using novel systems including a DTS (distributed temperature sensing) and DAS (distributed acoustic sensing) at the edge of a medium-size marine terminating glacier in southwest Greenland. The power and novelty of these systems is the ability to record variability on a wide range of scales (from milli-seconds to days) generated throughout the entire cycle of icebergs formation and drift: initial fracturing, detachment and subsequent drift away from the calving front. The authors convincingly describe the various types of waves (or wave-like processes) associated with these events including seismic waves from the fracturing, waves excited at the water/sediment interface (Scholte waves), tsunamis (transient surface gravity waves) excited by iceberg calving, and internal gravity waves in the fjord – including those present in the wake of the iceberg. Through several key examples for each type of waves, they show that the combined DTS and DAS systems can successfully record these waves and provide information on their propagation, amplitude and origin location.

Given the ability of these systems to record both the calving and the waves generated by calving events – the authors then speculate that these systems could be used to track calving events, separate aerial and submarine calving events, quantify calving volumes, advance detection of tsunami waves and quantify the impact of internal gravity waves on mixing and submarine melting. Furthermore, they conclude that this study provides evidence of a potential feedback where calving excites waves, which increase melting at the ice/ocean boundary, which then impacts calving.

The novelty of the paper is that it demonstrates that DTS and DAS sensors can be successfully deployed in front of calving glaciers and that, when deployed, they record a multitude of transient wave-systems generated throughout the calving cycle. Some of these processes had already been recorded using different observing techniques (eg surface gravity waves and location, Minowa et al. 2018, or internal gravity waves, Cusack et al. 2023) – the novelty here is in demonstrating that they can all be recorded by this system, including over larger distances and at much higher resolution/information. From a science result perspective, showing that many of these processes are occurring (and hence interacting) at the same time is novel. Some of the sediment/ocean waves and the impact of waves on temperatures at the seafloor are also novel. The observations and analysis techniques described here have the potential to revolutionize the study of the glacier/ocean interface.

What this study does not do is present new findings that use the observations made to quantitatively assess the impact of e.g. one particular wave-type on a glacial process. For example, the authors use vertical velocity data, together with other fjord collected observations, to show how submarine melting could be estimated but admit that more data and more analysis would be needed for a more realistic estimate. Similarly, they argue that these systems may be useful in early detection of tsunamis, generated by calving, which may pose a threat to people living in the vicinity of the glacier. But much more work and planning would be required to determine the feasibility of such a system.

The paper is well written, the analysis presented is sufficiently detailed and explained. In summary, the strength of this study is in the novelty of the methodology and a demonstration that it can record a wide range of wave-types associated with calving processes. This opens the door to the deployment of this system in different glacial environments to quantify and describe these different processes in greater detail. As such it is an extremely valuable step forward in our understanding of how to observe these challenging systems.

We thank referee #2 and referee #3 for their review. Even though the referees do not make specific suggestions for revisions, we still would like to make three minor clarifications here:

First, we would like to emphasize, that Cusack et al. 2023 do not link their observation of internal gravity waves to distinct calving events or calving activity in general, but to subglacial discharge. In contrast, our

result link individual internal gravity wave trains to particular calving events. This does not mean that our observation rules out the validity of the interpretation from Cusack et al. 2023.

Second, we would like to point out that we do include quantitative assessments where our data allows for them. This is the case for the submarine melt calculation from calving-induced internal wave activity. However, we want to make clear that our DAS/DTS records do not provide direct observations of submarine melt. Regarding other quantifications, we now additionally included a subaerial frontal ablation budget based on calving event statistics that shows that DAS may observe as much as 98% of all ice volume loss through subaerial calving events.

Third, we would like to mention that DAS systems have been deployed for the purpose of tsunami early warning, e.g. at Mount Ätna (by GEOMAR, Kiel and GFZ, Potsdam). From our DAS dataset, tsunamis can be triggered minutes before they reach the shore. However, drifting icebergs that cause the DAS data to saturate, reduce the reliability of such an automatic trigger.

With these minor clarifications, we appreciate the referees' assessment of our manuscript as an 'extremely valuable step forward'. Thank you.

Comments of Referee #4 and Response:

Dear Nature Editors and Manuscript Authors,

In this study, Gräff et al describe observations conducted near a major south Greenland glacier (Eqalorutsit Kagilliit Sermiat) using a fiber optic cable (and other instruments). Their measurements detect many of the processes known to cause ice loss from tidewater glaciers (e.g. “subaerial” calving, internal waves) as well as some processes that have been largely unobserved (e.g. submarine calving, iceberg currents). The observations reveal a “chain” of processes that contribute to ice loss from the glacier. This study is timely and of scientific interest because frontal ablation-induced ice loss from Greenland’s glaciers leads to a substantial portion of mass loss from the ice sheet and global sea level rise.

The use of the fiber optic cable for this application is extremely novel and the integrated observations from this field campaign are likely to be of widespread interest to the glaciological and oceanographic communities and beyond. In reading this manuscript, I had two major comments described here as well as a number of line and figure suggestions, appended below.

First, given that the fiber optic cable is the novel new observational instrument in this study, there is a lot of missing information that the reader needs to interpret how this instrument works and how the authors have made their conclusions. As of now, there is a major leap in logic that is hard to follow. One main update that would help readers is a paragraph circa line 112 that describes how the fiber optical cable is sensing changes in the water (e.g. is it expanding and contracting? something else? what is a channel? why is phase important? etc) and how these signals are sensed, parsed, and eventually interpreted after collection. I don’t think the paragraph should be highly technical (although perhaps some more technical details summarizing the measurements should be placed in the methods) but some sort of description is necessary. This would also really help the reader in interpreting many of the main figures (e.g. 2 and 3) which are currently lacking in units and even colorbars, obscuring a clear understanding. I get the sense that the authors may have looked at these types of figures long enough that they are easily interpretable to them but that is not the case for the standard interested scientist.

We understand the referee’s concerns and would like to elaborate on them:

First, we understand but do not entirely share the referee’s opinion that ‘the reader needs to interpret how this instrument works’. In our opinion, understanding the complex functionality of DAS and DTS is not necessary to comprehend the conclusions of this manuscript. Our argument is that the reader also does not need to understand how the CMOS sensors in our cameras work to see calving events in the images. The same applies to all other sensors that we used. We therefore also do not share the opinion that ‘there is a major leap in logic’.

Previously, we had the measurement principles of DAS and DTS included in the Methods section. However, we can understand that the interested reader wants to know how quantities like temperature and strain can be measured spatially resolved along an optical fiber without looking into the Methods section. Therefore, we now tried to condense this information now into four sentences that we included as the referee suggested upfront in the introductory text:

“Distributed Acoustic Sensing (DAS) and Distributed Temperature Sensing (DTS) interrogators convert optical fibers into linear arrays of thousands of seismo-acoustic and temperature sensors along fiber-optic cables. The DAS technology is based on measuring phase changes in the Rayleigh-backscattered laser light, which can represent both variations in strain and the index of refraction in the fiber and may be caused by vibrations, pressure- and temperature fluctuations. In contrast, DTS measures the intensity of Raman-backscattered laser light from inelastic interactions between temperature dependent molecular vibrations within the fiber, allowing to determine strain independent absolute temperature (Methods). For both DAS and DTS, the time of flight between laser light emission and the detection of the backscattered light determines the sensing locations along the fiber which are discretized and referred to as ‘channels’. Real-time data telemetry from these channels to the recording units, called ‘interrogators’, is established

by the traveling laser pulses. This allows for deployments in harsh cryospheric environments, while the interrogators can remain housed in more hospitable locations.”

More technical aspects of distributed fiber sensing can now be found in the Methods (formerly in the Supplementary Information), to which we refer to.

We can also understand the concerns regarding the interpretation of our figures, and we are aware that they are different from standard line-style plots that are most abundant in scientific literature. We now made it clearer in all figure captions what is shown in the plots (specifically for the DAS data).

Second, one of the main claims and selling points of the paper is that the observations from the fiber optical cable help “close the ice front ablation budget” (Line 72) and indeed a lot of novel processes are identified. However, the ice front ablation budget is never explicitly calculated. This calculation would significantly improve the impact of the manuscript and would help make the case that this type of instrumentation is a breakthrough in assessing ice sheet mass balance. Thus, I think it is necessary to attempt a budget calculation, even if the uncertainties are large, so that readers can understand what processes are “important” and where focus should be placed for future studies. This calculation would also make this paper a blueprint for future studies with fiber optic cables near glaciers. In addition, it would also help support the authors expectation that this will be applied in diverse studies at other fjord systems (line 353).

We can understand the referee’s suggestion to calculate the ablation budget explicitly, and obviously this would be most relevant for the submarine part of the glacier terminus. Although we support this idea, our data unfortunately does not allow for a solid calculation and stated quantities would be misleading. We refer to the guidance of the editor in this regard and list reasons for our decision below.

However, as we acknowledge the referee’s comment, we now quantify the subaerial ablation budget in the ‘Iceberg detachment’ section: “... *our seafloor DAS resolves and locates ten-thousands of iceberg detachments as small as $\sim 100\text{m}^3$. This is much below the detection threshold of most other methods such as remote sensing platforms, local seismometer networks, timelapse imagery and terrestrial interferometric radar (TRI) scans. Consequently, our DAS-based calving catalog comprises 200x more detections and a 40% larger cumulative detachment volume than the TRI detected (see Methods, Extended Data Fig. 4)*”

We furthermore add details about this calculation to the Methods section: “Comparing the ice flow based on a satellite derived mean ice-flow velocity across the calving front of 4.5 m/s, a front width of $\sim 3,500$ m and an average height of ~ 80 m, and accounting for changes in the front position, the TRI detects only $\sim 35\%$ of the actual calved-off ice volume. However, due to smoothing, 50% of the measured volume may get lost in the TRI detection process, increasing the TRI sensitivity up to $\sim 70\%$ of the actual calved-off volume. Now considering the estimated cumulative calving volume detected with DAS is 40% larger than with the TRI, this means that DAS may detect up to $\sim 98\%$ of all solid frontal ablation. Note that unlike the TRI, DAS is also able to detect submarine calving events.”

Reasons for why we do not explicitly calculate a frontal submarine ablation budget:

1. Our terrestrial radar interferometer (TRI) measurements of ice flow and subaerial calving does only overlap a few days with our DAS/DTS observations of internal wave activity, resulting in a short observation period.
 - ➔ Therefore, a single large calving event (full glacier height, multiple ten-meter thickness) will distort the relative contributions of calving and submarine melt (on the order of meters per day), when calculating an ablation budget.
2. The TRI does not measure submarine calving. With our DAS measurements we are not able yet to quantify submarine calving volume explicitly. We can only make a rough estimate what the total calving volume recorded with DAS may be.
 - ➔ Therefore, when calculating an ablation budget, all submarine mass loss would be accounted to submarine melt, which is wrong as we observe submarine calving by eye and in the data. We just cannot quantify the volume.

3. We do not measure submarine melt. Modelling the submarine melt based on subglacial discharge and fjord circulation would be out of the scope of this study and we would not be able to constrain the model with measurements
→ Therefore, this would not add any novelty or knowledge to our study.

We now address the referee's point "so that readers can understand what processes are 'important' and where focus should be placed for future studies" by explicitly stating this in the second-last sentence of the main text: *"This result potentially explains why certain submarine melt models underestimate submarine melt by up to two orders of magnitude and should be investigated further."*

Overall, I recommend this paper for publication in Nature after addressing the major comments above. Please don't hesitate to reach out to me if you have any questions about this review.

Sincerely,

Mike Wood

Line comments:

- 2: I might suggest "revealed" in place of "illuminated" since the cable is not an active (vs passive) sensor (...right?)
Good point. It will probably be replaced by "resolved": *"Calving-driven fjord dynamics resolved by seafloor fiber sensing"*
- 82: Should probably define GrIS here.
Done.
- 94: strongly affect calving activity. > strongly inhibits calving in winter in systems with especially thick mélange.
We rewrite to: *"... strongly inhibit calving activity."*, as we mention *"tightly packed"* before and the impact of mélange on calving should not be limited to winter.
- 97: re: thermal structure: Davison et al 2020 show the effect of icebergs on thermal structure so this is not totally "hidden"
True, Davison et al 2020 predicts the impact of icebergs on the thermal structure, but the experimental proof is still missing. However, we agree that there are hints (also in our data) that this theory is correct.
- 112: I might also include being hidden at depth away from icebergs as a benefit
Our cable got destroyed by icebergs at multiple occasions. Therefore, we do not really want to emphasize this as a benefit.
- 117: 0.3% of the GrIS area
Good point! The area, not the volume! Done.
- 145-146: This is a little confusing – are there not "wiggles" in the plots above Figure 2 at the same time as the wave arrival? Units and a description of these subplots in Figure 2 would be helpful (see comments on figures below).
Yes, there are wiggles. However, we do not know what kind of wiggles, i.e. what wave type traveling at what speed and with what characteristics (dispersive, non-dispersive) along what wave path (refracted, reflected, direct, ...). In the DAS data, this is immediately clear. To make it clearer, we added: *"lack interpretable phase arrivals (Fig. 3a/b upper panels) – therefore impeding precise event location, the DAS record sections..."*
For subplot units, see Figure comments.
- 150: Why would water-sediment waves influence ice front stability? This seems speculative without an explanation.

We agree. This is not written clear enough. The observation is that thick sediments (multiple 100m thick are present. It is likely that the glacier is also bedded on these sediments. If not, a multiple hundred-meter bedrock step at the calving front would need to be present, which seems unlikely to us. Thick sediments under ice streams are likely responsible for fast ice flow and play a crucial role in the tidewater glacier cycle. We reformulated: "... indicate a thick sedimentary cover that likely affects ice front stability and ice stream dynamics."

- 160: moment -> timing
Timing is not really what we mean. We literally mean the "moment".
- 168: remove "into"
Done.
- 172-173: If submarine calving events are not attached to the sea floor or fjord walls (and therefore do not provide buttressing) then do they really matter in models for long term evolution of glaciers and ice sheets? If so, this needs an explanation.
Submarine calving is simply an ice loss mechanism, just as subaerial calving. Limiting calving to only subaerial would ignore up to 90% of the ice column.
- 180-181: probably need a citation here
Yes, added: Kugler, S., Bohlen, T., Forbriger, T., Bussat, S. & Klein, G. Scholte-wave tomography for shallow-water marine sediments. *Geophys. J. Int.* **168**, 551–570 (2007).
- 188-189: former -> latter? Submarine detachments would cause a cavity, correct?
No, subaerial calving events do as they fall into the water. Just as an air cavity that forms from diving into the water.
- 227: vertices? (epicenters is plural)
Yes. Changed.
- 239: What is the depth of the internal waves/thermocline? This will dictate the extent to which they impact melt. (e.g. are the IGW-induced temperature variations and water velocities mostly near the surface? Or at depth where the glacier is more vulnerable?)
This changes over time and can be seen from Fig. 5a. We added: "... periods varying between ~15-30 min at various depths (Fig. 5)"
- 242: vertical water velocities?
Both vertical and horizontal.
- 249-251: There is a lot of jargon in these lines for the general reader (Kelvin-Helmholtz instabilities, Turner angles). Could this be said a different way here with some of the details put into Methods for the interested reader? I see Turner angles are in the Methods already.
This is a valid concern. We removed the term '*Kelvin-Helmholtz instabilities*', as it only puts a term on shear-driven mixing and does not add extra information. We also reformulated the sentences about the Turner angle to make it more accessible.
- 264: breaking
Done.
- 264-265: This seems a little speculative without additional information
Yes, we reformulated: "*Additionally, transient small-scale overturnings are visible, where warmer water temporarily is located above colder water (e.g., 28 Aug 0:00 around 2975m OD and Fig 5b), possibly formed by subglacial discharge or meltwater intrusions or breaking internal waves. These overturnings likely cause a regime change from diffusive convection to salt fingering,...*"
We also strengthened the formulation that these overturnings likely cause a regime change, as indicated by the calculated Turner angles. "*This mechanism requires low current speeds and the absence of shear-driven mixing. Turner angles, as a measure of the water column's stability regime, are calculated to $Tu = -50^{\circ}6^{\circ}$ below the glacially modified waters at ~0-100m depth, i.e. subglacial discharge mixed with oceanic water (Methods, Extended Data Fig. 6a, Fig.7) confirm that diffusive convection is active.*"
- 271: to calving events -> as originating from calving events
Done.
- 297: what is a "hyperbolic move out"?
We replaced with '*arrivals*' to make it clearer as 'move out' is seismology jargon.

- 301: illuminate -> reveal (same comment above)
Done.
- 301-304: Do I understand this section right that an internal wave travels back toward the glacier from the calved iceberg that is floating away? If so, it would be good to explicitly mention this.
No. The IGW wake travels behind the drifting iceberg. As iceberg drift initiates at the glacier, the wave wake also forms there. We clarify by: "... and—as a source of kinetic energy originating at the glacier terminus—likely increases the ice front ablation."
- 337-338: I think this needs to be qualified/nuanced a little. Ice mélange is shown to contribute to buttressing during winter at very large glaciers with thick, jam-packed icebergs in a fjord. I doubt this process is at play at EKs.
We have evidence that it plays a role at EKs. Fig. R2 attached to this document shows a picture taken in March 2024. The mélange is buttressing in an extent that it impedes two large icebergs to detach completely from the glacier. To clarify we added: "*which has been observed to buttress the ice-front and suppress iceberg calving when sufficiently compact.*"
- 341: remove bespoke
Done.
- 352: what is a wide spatial aperture and why is it important (not mentioned until here in the conclusions)
The spatial aperture is the extent of the sensors, i.e. the cable. We replaced the term with 'coverage'. It is not as precise, but more accessible.
- 373: the IGWs are induced by calving but calving is most prominent in the melt season – why would it be most important outside the melt season?
We agree that the formulation is not precise enough. The relative importance is high in winter because the subglacial discharge is minimal. We reformulated: "*This multiplier effect is therefore expected to have the largest relative importance outside the melt season, when strong subglacial discharge plumes are lacking, and submarine melt in traditional theories and models would be suppressed.*"
- 375: Were there observed major calving events in the Sutherland et al study between the front scans that led to this 2 orders of magnitude conclusion?
Yes, during the two observation periods of Sutherland et al. 2011: "Extensive retreat and calving were observed on the southern terminus in both seasons." See their SI Fig. S9. We can only speculate, if the calving activity in that study and the induced internal gravity wave (IGW) activity can explain the observed submarine melt of up to 8 m/day in August. However, given the high water temperatures measured in August (6-8°C) and the strong stratification, combined with ice flow speeds of up to 30m/day, an on average constant calving front location and the resulting calving activity, we assess the calving-induced IGW activity to affect submarine melt significantly.

Figure Comments:

- Figure 1: Red rectangle -> Green rectangle? Also, I think "orthophoto" should be "imagery".
Misunderstanding. Done.
- Figure 2: This figure is unclear because it is missing a lot of information. Some questions:
 - What are the units on the top panels?
We included the axis label and the units.
 - What are the colors in the plots (blues and reds)?
Very good point. We now include: "*Blue indicates compressional, red extensional strain.*"
 - What are the units?
See comments on the units of the DAS data for Referee #1. The DAS data is in units of strain normalized for each channel. We now emphasize this in the caption.
 - What are the horizontal variations around 2000 and 2600 m?
Great question. They are continuous vortex-induced-vibrations. In this context, they can be regarded as noise. Therefore, we do not explain it explicitly.

- If QJT01 is closer to the Bay, then why does it receive a signal before QJT02, which is closer to the calving event?
Probably a misreading. Fig 1b shows the glacier fracturing event as a red star at the calving front. It is closest to QJT01.
 - Also, it's difficult to see red and blue dots on a red and blue figure.
Fixed.
- Figure 3: Suggest labeling calving events explicitly with annotations and text
We added circles in the top panel that mark every calving event.
- Figure 4: This figure is also unclear.
 - What is a "DAS 50m stack"?
Tried to clarify by changing to: "...alongside the mean DAS signal over the 50 m (10 channels) cable section closest to the tide gauge"
 - What do the colors indicate in panels b and c?
Good point. We added a color bar. It is power spectral density.
 - What are the units? Are they the same (if not, perhaps a different colormap for each would be a good idea)?
Done. dB ms and dB/Hz.
- Figure 5: Some selectively chosen isotherms on panel a would help the reader identify which features are important in this figure.
We really tried to plot isotherm following the 'thermoclines', as we know that oceanographers love them. However, we decided to not include them for the following reasons: 1) The temperature is not monotonically increasing with depth due to noise in the measurement, overturnings, undersampled internal waves, etc. 2) Due to DTS calibration artifacts, the temperature at which the internal waves are most dominant (Fig. 5 yellow to orange to purple to blue) slightly varies. 3) Adding meaningful isotherm contours would require so much smoothing, that the internal waves would not be visible anymore. Fig. R3 attached to this document shows an example with isotherms at 0.5°C, 1.2°C, and 1.8°C as contour lines.
 - Also, what is happening in the transition just after 2023/8/27 21:00?
That is the overturning. We added an annotation.
- Figure 6:
 - What is the temperature anomaly in reference to?
We added: "*DTS deep seafloor temperature anomaly relative to the median temperature during the passage of two IGW wave wakes ...*"
 - What is the difference between b and d (just the scale)?
Explained in the figure caption.
 - What is a DAS Channel (not described in text, see first major comment).
We define it now in the third paragraph of the main text: "the sensing locations along the fiber which are discretized and referred to as 'channels'."
- Figure 7: This is a nice schematic and I feel like it should come earlier in the manuscript. I realize all the text builds up to this figure but since it encapsulates the "story" perhaps it could be moved sooner? Also, since this is a "chain" of events, it might make better sense for these to be labeled with numbers (Step 1, Step 2, ...) rather than letters (which are used for subplots in the previous images).
Moved upfront.

Additional Figures:

Figure R1: Ray tracing of tsunami waves. At the tide gauge location (easternmost cable section), the waves arrive oblique and therefore hindering a dispersion analysis.

Figure R2: Picture of the calving front of EKaS taken in March 2025. The *mélange* impedes large parts of the glacier to detach completely. Credits: Raphael Moser

Figure R3: Demonstration of including isotherms as contours. Even with applying Gaussian-filtering in space and time, the contours are rather hiding the IGW oscillations. We therefore would like to refrain from plotting them.

Figure R4: Tsunami trigger on tide gauge and DAS data. For the shown DAS representation, a 10-channel stack is used.

Bibliography:

1. Walter, F. *et al.* Distributed acoustic sensing of microseismic sources and wave propagation in glaciated terrain. *Nat. Commun.* **11**, 2436 (2020).
2. Booth, A. D. *et al.* Distributed acoustic sensing of seismic properties in a borehole drilled on a fast-flowing Greenlandic outlet glacier. *Geophys. Res. Lett.* **47**, e2020GL088148 (2020).
3. Fichtner, A. *et al.* Fiber-Optic Airplane Seismology on the Northeast Greenland Ice Stream. *Seism. Rec.* **3**, 125–133 (2023).
4. Hudson, T. S. *et al.* Distributed Acoustic Sensing (DAS) for natural microseismicity studies: A case study from Antarctica. *J. Geophys. Res. Solid Earth* **126**, e2020JB021493 (2021).
5. Klaasen, S., Paitz, P., Lindner, N., Dettmer, J. & Fichtner, A. Distributed Acoustic Sensing in Volcano-Glacial Environments—Mount Meager, British Columbia. *J. Geophys. Res. Solid Earth* **126**, (2021).
6. Manos, J.-M. *et al.* DAS to discharge: using distributed acoustic sensing (DAS) to infer glacier runoff. *J. Glaciol.* **70**, e67 (2024).

Comments of Referee #1 and Response:

The authors have addressed most of our and others' first review round of comments. The revised manuscript is slightly more concise, and the changes to both the figures and the text have improved the readability of the article up to a point. The authors have made efforts to quantify more of their observations (example L295-298), which is helpful. However, the track changes document mostly highlights the minor changes due to citation reordering and figure reordering. There are only a few substantial modifications to the text that in our opinion reflect how the constructive comments and suggestions given by the 4 reviewers were addressed.

We thank referee #1 for the feedback. Referee #4's recent comment captures what we were able to do within the constraints of our data: "The authors [...] provide an estimate of terms that are possible to robustly quantify with their measurements. In addition, they have convincingly described some of the reasons why a full closure of the budget is not possible with their sensors in the response to reviewers."

Some positive feedback is that we find moving the illustration that describes the processes as Figure 2 helps. However, the text frequently refers to figures out of sequential order, which makes it hard for the reader to go back and forth. We are afraid we do not have any new useful comment on this, but a more logical and consistent way to align text and figure order would further enhance the clarity.

We agree that changing Fig. 2 from its purpose of a concluding and summarizing figures to an overview figure requires the reader now to go forth and back. As in the first review round comments towards more in-text-references to the figures were made, we would like to keep the references to Fig. 2.

Minor/moderate points:

- L 125: Consistent use of fiber and fibre?
Thanks, we stick with 'fiber' in the American English spelling.
- L 156: Would 'travel times' be a better fit rather than 'time of flight'?
Yes. Changed.
- L 190: There is no blue curve on Figure 3b – perhaps the authors mean 3d.
Thanks. That was indeed a typo. Changed to Fig. 3d.
- L 279: Period missing.
Inserted.
- L 391: "Submarine melt" appears twice in the same sentence. Maybe replacing "submarine melt models" with glacier models would be better phrasing.
Good point. We changed to 'calving front models', as it is more precise.
- Figure 4 caption (previous reviews):
The authors misunderstood 'define'. Please describe/note/quantify the number that represents the "deep water limit" in the caption. We did not request the equation, just a number to help the reader understand the figure.
Thank you for the clarification. The deep-water limit is of course frequency- (or better wavelength-) dependent. To help the reader understand the figure, we now included: "(>30 m for relevant frequencies)"
- Regarding the pinning point island comment – that is why we asked. It is not in the caption, yet it is referred to in the text when describing Figure 4. Please add or modify to clarify.
Ok, thanks for clarifying. We added an annotation in Fig. 4e.
- Figure 6(e). This doesn't seem important enough to be shown in the main text. Perhaps consider moving it to the Extended Data.
In this regard, we disagree. Fig 6e shows the length of suspended cable sections and the coherent oscillations. The frequency of the narrow-banded tension dominated modes in Fig 6f is controlled by this length. These modes in turn, are the core of our observations as they get excited when the vortex shedding frequency passes their frequency range. We emphasized this now in the main text: "*From this, narrow-banded tension dominated natural frequencies of the cable get excited that scale inversely with the length of the suspended cable section. The resulting spectral signature allows for measurements of current speed perpendicular to the fiber-optic cable and perpendicular to the calving front (Fig. 6f)*"

Comments of Referee #2 and Response:

I co-reviewed this manuscript with one of the reviewers who provided the listed reports.
We also thank referee #2 for their contribution.

Comments of Referee #4 and Response:

Dear Nature Editors and Manuscript Authors,

This is my second review of Gräff et al's manuscript describing observations conducted near a major south Greenland glacier (Eqalorutsit Kagilliit Sermiat) using a fiber optic cable and other instruments. In response to comments from me and the other reviewers, this manuscript has undergone significant revisions that improve the overall presentation of the study.

In my previous review, I had two major comments, which I outline here along with my summary and impressions the authors' response.

My first comment pertained to the description of the DAS/DTS systems and a lack of introductory information on how the system works. The authors have now provided 4 sentences in their introduction in the manuscript about this system which greatly improves the reader understanding of what the system is measuring and what some of the jargon means. This is sufficient to address the point I brought up (and I certainly didn't intend for a full description of the physics of the measurement system at this point in the manuscript). I also noted in this comment that many of the plots were missing colorbars. Some colorbars were added to a few plots and the authors have elected to describe the blue/red compressional/extensional strain in the caption of figures without colorbars.

My second comment pertained to closing the ablation budget, which was not quantified yet was indicated as an important outcome from the measurements presented in this study. The authors have now updated the manuscript to provide an estimate of terms that are possible to robustly quantify with their measurements. In addition, they have convincingly described some of the reasons why a full closure of the budget is not possible with their sensors in the response to reviewers.

In addition to the major comments above, I also provided several line and figure comments and suggestions. The authors have gone through each of the suggestions and provided, where possible, appropriate updates to the manuscript and/or responses to each of these points.

Overall, my comments have been addressed and I recommend this manuscript for publication in Nature.

Sincerely,

Mike Wood

We thank the referee for the concluding report and the comments made in the first review round, which – also in our eyes– improved the manuscript.